

# The recent state and variability of the carbonate system of the Canadian Arctic in the context of ocean acidification

Alexis Beaupré-Laperrière[1], Alfonso Mucci[1] and Helmuth Thomas[2]

[1]GEOTOP and Department of Earth and Planetary Sciences, McGill University, 3450 University Street, Montreal, Quebec, H3A 0E8, Canada

[2]Dalhousie University, Department of Oceanography, Halifax, NS, Canada (now at Center for Materials and Coastal Research, Helmholtz-Zentrum Geesthacht, Germany)

*Correspondence to*: Alexis Beaupré-Laperrière (alexis.beauprelaperriere@mail.mcgill.ca)

## Abstract

Ocean acidification driven by the uptake of anthropogenic $CO_2$ by the surface oceans constitutes a potential threat to the health of marine ecosystems around the globe. The Arctic Ocean is particularly vulnerable to acidification due to its relatively low buffering capacity and, thus, is an ideal region to study the progression and effects of acidification before they become globally widespread. The appearance of undersaturated surface waters with respect to the carbonate mineral aragonite ($\Omega_A < 1$), an important threshold beyond which the calcification and growth of some marine organisms might be hindered, has recently been documented in the Canada Basin and adjacent Canadian Arctic Archipelago. Nonetheless, few of these observations were made in the last five years and the spatial coverage in the latter region is poor. Additionally, the strong variability inherent to this dynamic shelf environment renders the temporal imprint of ocean acidification on carbonate system parameters (pH, $pCO_2$, DIC, $\Omega$) virtually indistinguishable on decadal timescales. We use a dataset of carbonate system parameters measured in Canadian Arctic Archipelago (CAA) and its adjacent basins to describe the recent state of these parameters across the Canadian Arctic and investigate the amplitude and sources of the system's variability. Our findings reveal that, in addition to the surface of the Canada Basin, the entire water column of the Queen Maud Gulf was undersaturated with respect to aragonite in 2015 and 2016. We also estimate that approximately a third of the interannual variability in surface DIC in the CAA results from fluctuations in biological activity.





## 1 Introduction

Ocean acidification and its repercussions on marine ecosystems constitute an important consequence of the ongoing rise in atmospheric carbon dioxide ($CO_2$) concentrations. The world's oceans absorbed approximately a third of the anthropogenic $CO_2$ released to the atmosphere over the last two centuries of industrial activity (Sabine et al., 2004) and are currently a sink for ~24% of global annual anthropogenic carbon emissions (Le Quéré et al., 2018). Atmospheric carbon dioxide uptake by the surface oceans has well defined impacts on seawater chemistry, including a decrease of pH resulting from the dissociation of carbonic acid ($H_2CO_3$), the product of the reaction between water and dissolved $CO_2$. A large fraction of the hydrogen ions released by this reaction is neutralized by carbonate ions ($CO_3^{2-}$), leading to a decrease of their concentration and, concomitantly, the saturation state of seawater with respect to the carbonate minerals calcite and aragonite. The saturation state is defined by:

(1) $\quad \Omega_{C,A} = [Ca^{2+}][CO_3^{2-}] / K^*_{SP}$

where square brackets denote concentrations and $K^*_{SP}$ is the stoichiometric solubility product of calcite or aragonite, the two most common marine $CaCO_3$ polymorphs, at a given temperature, pressure and salinity.

The combination of these phenomena is most often referred to as ocean acidification (OA). As it proceeds, the dissolved inorganic carbon concentration (DIC; the sum of $[H_2CO_3^*]$, $[HCO_3^-]$ and $[CO_3^{2-}]$) in the surface ocean is expected to increase relative to the total alkalinity (TA; the capacity of a solution to neutralize protons), as the latter is nearly conservative in the surface ocean (Wolf-Gladrow et al., 2007). The global mean surface ocean pH currently sits ~0.1 units below its preindustrial value (Orr et al., 2005) and, according to Earth System models, under the IPCC's "business as usual" RCP8.5 emission scenario, is predicted to decrease by an additional 0.3 units by the end of this century (Bopp et al., 2013).

Marine calcifying organisms, many of which are important primary producers (e.g., coccolithophores), extract the constituents of their calcitic or aragonitic tests (shells) from seawater. In most cases, their ability to do so is directly dependent on the saturation state of the



surrounding water. Supersaturated seawater ($\Omega > 1$) will favor carbonate precipitation whereas undersaturated seawater ($\Omega < 1$) favors carbonate dissolution. The majority of calcifying organisms, such as planktonic foraminifera and coccolithophores, undergo dissolution or exhibit substantially hindered growth when exposed to undersaturated seawater (e.g., Mostofa et al., 2016). Calcifying organisms found in the Arctic Ocean are subject to rapid environmental changes,

as this polar ocean is warming more rapidly than others (Serreze et al., 2011) and is particularly vulnerable to acidification due to the low alkalinity and correspondingly weak buffer capacity of its cold waters (Shadwick et al., 2013). Atmospheric $CO_2$ uptake by Arctic surface waters is further promoted by the rapidly melting seasonal ice cover (e.g., Stroeve et al., 2012), which exposes a gradually larger area of the ocean to gas exchange with the atmosphere and whose melt-water

dilutes calcium concentrations, alkalinity and carbonate ion concentrations, further decreasing $\Omega$.

         The Canadian Arctic Archipelago (CAA) and its adjacent deep basins, the Canada Basin (CB) and Baffin Bay (BB, Fig. 1), are part of the region projected to undergo the largest reduction in ice cover and, consequently, the largest decrease in surface pH (~0.6) and $\Omega$ (1 and 0.7 for $\Omega_C$

and $\Omega_A$, respectively) over the 21$^{st}$ century (Popova et al., 2014). Recent observations (e.g. Yamamoto-Kawai et al., 2009b; Robbins et al., 2013; Qi et al., 2017) hint at a significant decrease of the aragonite saturation state of surface waters as well as a rapid expansion of the undersaturated area in the Canada Basin, but most of these time series predate 2010 and do not extend geographically to the CAA and Baffin Bay.


         In this study, we use a large observational dataset for this part of the Arctic to 1) describe the recent state of the carbonate chemistry and its spatial variability in the Canadian Arctic Archipelago and the adjacent basins, 2) investigate the interannual variability in carbonate system parameters and identify detectable temporal trends using time series spanning from 2003 to 2016

and 3) estimate the contribution of the temporal change in biological activity to the observed variability of surface DIC.





## 2 Study area

### 2.1 Canada Basin


The Canada Basin (CB), Canadian Arctic Archipelago and Baffin Bay accommodate the flow of surface waters from the North Pacific to the North Atlantic (Stigebrandt, 1984), as well as circulation of Atlantic waters at greater depths. The water mass structure of the southern Canada Basin is representative of these broad circulation patterns and can be summarized as follows

(Carmack et al., 1989; MacDonald et al., 1989; Lansard et al., 2012): a relatively cold and fresh surface layer that contains significant fractions of meteoric water (river discharge and precipitation) and sea-ice melt in the summer and becomes homogeneous in winter; an intermediate layer (~50-200m) of advected Pacific waters, often divided into summer and winter varieties, the latter being distinctively rich in nutrients and metabolic $CO_2$ and recognizable by a

temperature minimum in the upper halocline; a layer of warm (~0.5°C) and saline ($S_P>34$) Atlantic water; a cold bottom layer with practical salinities ($S_P$) reaching 34.85. The main surface circulation feature in this area, the clockwise Beaufort Gyre, is the largest freshwater reservoir in the northern oceans, formed through Ekman pumping (Proshutinsky et al., 2009). This feature is reversed at depth. The main source of freshwater to the Beaufort Sea (the southwest portion of the

Canada Basin) is the Mackenzie River (Carmack and MacDonald, 2002), although the contribution of sea-ice melt is significantly increasing along with the accelerating reduction in ice cover (Yamamoto-Kawai et al., 2009a). The supply of freshwater at the surface, combined with the advection of pre-acidified waters from the Pacific (100-200 m) and the Atlantic (below 400 m; Luo et al., 2016) Oceans result in the presence of three distinct and expanding undersaturation

horizons in the Canada Basin (Wynn et al., 2016).

### 2.2 Canadian Arctic Archipelago

The CAA is a series of islands on the Canadian continental shelf, through which complex circulation patterns unfold in narrow and relatively shallow channels (<500 m). See McLaughlin

et al. (2004) for a detailed description of the circulation and water mass processes in this area. The prominent pathway for the eastward flow of Canada Basin water masses into Baffin Bay extends along the 74[th] parallel from M'Clure Strait to Lancaster Sound. A shallow 125-m sill at Barrow



Strait, located centrally in the archipelago, inhibits the eastward flow of Atlantic waters, so that only surface and Pacific-origin waters reach Baffin Bay (Bidleman et al., 2007). The properties of

these water masses are substantially modified during this transit (McLaughlin et al., 2004). East of Barrow Strait, Atlantic waters originating from the Labrador Sea penetrate the archipelago through Baffin Bay. Smaller inflows of water from the deep Arctic Ocean into the archipelago occur through the Queen Elizabeth Islands (North-East) and Nares Strait; minor outflows occur through Jones Sound and into Hudson Bay via Foxe Basin.  Notwithstanding the Mackenzie River,

whose discharge is limited to the Beaufort Sea and Amundsen Gulf, the southern portion of the CAA receives a considerable amount of freshwater from other large North American rivers (e.g., Coppermine, Burnside, Black).

### 2.3 Baffin Bay

The oceanographic regime of Baffin Bay is distinct from that of the CAA and CB, as it receives multiple inputs from both the Arctic and Atlantic Oceans. Cold and relatively fresh Arctic and Pacific-derived waters enter this 2300-m deep semi-enclosed basin through the Nares Strait as well as Jones and Lancaster Sounds (Muench, 1971; Jones et al., 1998, 2003). Warmer and more saline Atlantic Ocean waters are transported from the Labrador Sea by the West Greenland Current

(WGC) into Baffin Bay through the eastern side of Davis Strait, circulate cyclonically, i.e., in an anti-clockwise direction, before joining the southward Baffin Island Current (BIC) which exits Baffin Bay through the western Davis Strait (Bourke et al., 1989, Munchow et al., 2015). Atlantic Ocean waters are modified as they mix with Arctic inflows in Northern Baffin Bay, near the North Water Polynya (Melling et al., 2001). The resulting water mass structure is described by Tang et

al. (2004) as: 1) a cold (T<0°C) and relatively fresh ($S_P$<33.7) surface layer, representing the mixed Arctic inputs, 2) a warm (T>0°C) and saline ($S_P$>34) Atlantic Ocean water layer found at depths of ~300 to 800 m, and 3) a deeper layer of nearly constant salinity ($S_P$ = 34.5).




## 3 Methods

The dataset used in this study comprises data from 420 stations visited during various research cruises carried out aboard the CCGS Amundsen between 2003 and 2016. Table 1 summarizes the timeframe and relevant data acquired during each cruise; figure 2 shows the position of each sampling station. Although ice conditions restricted most observations to the summer months, two winter time-series, acquired in 2003-2004 (CASES, Miller et al., 2011) and 2007-2008 (CFL, Shadwick et al., 2011) are included in the dataset.

### 3.1 Sampling and measurements

Seawater was sampled separately for each measured parameter from Niskin bottles mounted on a Rosette system equipped with a Seabird SBE 911plus Conductivity-Temperature-Depth (CTD) sensor, which collected *in-situ* practical salinity ($S_P$) and temperature data throughout the water column. The conductivity/salinity probe was calibrated post-cruise against measurements carried out on discrete seawater samples using a Guildline Autosal 8400 salinometer (accuracy of $\pm$ 0.002 or less), itself calibrated with IAPSO standard seawater. Samples used for pH determination were drawn directly from the Niskin bottles into 125 mL low density polyethylene (LDPE) bottles with no headspace to avoid gas exchange with surrounding air and left to thermally equilibrate in a temperature bath set at 25.0 ($\pm$0.1) °C. pH, on the total proton scale ($pH_T$), was then measured spectrophotometrically on a Hewlett-Packard 8453 UV-visible diode array spectrophotometer using m-Cresol purple (Clayton and Byrne, 1993) and Phenol red (Robert-Baldo et al., 1985) indicators at 434 and 578 nm or 433 and 558 nm, respectively, in a 5-cm quartz cell. Daily calibrations were performed with TRIS buffer solutions at practical salinities of 25 and/or 35, depending on the salinity range of the samples. The reproducibility was found to be $\pm$0.005 pH units or better, based on duplicate measurements of the same samples with the same or both indicators. Samples destined for total alkalinity (TA) and dissolved inorganic carbon (DIC) analyses were drawn directly from the Niskin bottles into 250 or 500 mL glass bottles with ground-glass stoppers, poisoned with solid mercuric chloride ($HgCl_2$) to halt biological activity, and sealed with Apiezon M grease.





TA and DIC from the GEOTRACES 2009 and all post-2010 cruises were measured onboard or at Dalhousie University on a Marianda VINDTA 3C instrument, following the protocol described by Dickson et al. (2007) and calibrated with Certified Reference Materials (CRM) provided by A.G. Dickson (Scripps Institute of Oceanography). The precision of the instrument was found to be ± 2-3 µmol kg$^{-1}$ based on repeated CRM analyses. The remaining DIC and TA

analyses were performed respectively on a SOMMA instrument (Johnson et al., 1993) fitted to a UIC 5011 coulometer and by potentiometric titration using a Radiometer Titrilab 865 (see Mucci et al., 2010 for details), calibrated against the same CRMs used with the VINDTA.

        In-situ pH$_T$, DIC (ArcticNet 2009 only), pCO$_2$, $\Omega_C$ and $\Omega_A$, were calculated from TA and

pH$_T$(25°C) or TA and DIC (for 548 out of 4730 data points) with the Excel - 2.1 version of the CO2SYS algorithm (Lewis and Wallace, 1998), using the carbonic acid dissociation constants determined by Mehrbach et al. (1973), refit by Dickson and Millero, (1987), the HSO$_4^-$ dissociation constants of Dickson (1990) and the total boron concentration (B$_T$) from Uppström (1974). pCO$_2$ was normalized, using CO2SYS, to the mean temperature of the top 100 meters of the water

column (-0.4 °C). The resulting $\partial pCO_2/\partial T/pCO_2$ values range between 0.033 and 0.051 °C$^{-1}$, with a mean of 0.046 °C$^{-1}$, in good agreement with values obtained by Jiang et al. (2008; 0.027-0.042°C$^{-1}$) and Takahashi et al. (1993; 0.0423 °C$^{-1}$).

### 3.2 Quality control

In order to assess the robustness of the computed DIC values, we calculated DIC from TA and pH$_T$(25°C) and compared the results with the measured DIC values. The resulting coefficient of determination of the linear fit to the measured and calculated DIC values, R$^2$, is 0.989, while the mean difference between calculated and measured DIC values is ~2 µmol kg$^{-1}$. We excluded 30 measurements that differed from the calculated values by more than 50 µmol kg$^{-1}$ (2.5% of the

mean DIC).



Questionable TA measurements, excluded from the dataset, were identified as those outside a range of 3 standard deviations from the mean salinity-normalized TA for individual regions (CB, CAA, BB) characterized by internally consistent water mass assemblages. TA
measurements obtained from the two instrumental methods (VINDTA and Radiometer Titrilab 865) used in 2015 and 2016 were also compared to ensure that data originating from both methods could be used interchangeably in the calculation of additional parameters and conjointly in time series. The resulting coefficient of determination between both datasets ($R^2$) is 0.988, the mean of the non-systematic discrepancy between values is 6 µmol kg$^{-1}$ and its maximum is 36 µmol kg$^{-1}$,
respectively corresponding to 0.3% and 1.7% of the mean TA. The degree to which the results of this test are representative of the entire dataset is uncertain, but they constitute the best possible estimate of the uncertainty associated with the use of the two analytical methods used to measure TA. When TA measurements obtained from both methods deviated significantly ( >10 µmol kg$^{-1}$), specific alkalinity (TA/$S_P$), which should remain relatively constant in a given water mass of uniform salinity (Millero, 2005; p.268), was used to determine which data to discard. The deviation
from TA values calculated from DIC and pH$_T$(25°C) was used to complement the first method, especially at the surface, although the validity of DIC measurements was previously assessed using TA.

**3.3 Error estimation**

In order to quantify the error associated with the calculated carbonate system parameters reported in this study, we used the CO2SYS program modified by Orr et al. (2018), which applies error propagation to instrumental and constant-related uncertainties. For simplicity, we report the mean uncertainty for each parameter (see Table 2), as the variance is minimal within our dataset.
We found the additional uncertainty associated with the unavailability of nutrient concentrations (P and Si) as input parameters in CO2SYS to be negligible (up to 0.0006 pH units, 1.5 µatm pCO$_2$ and 0.006 Ω units, as determined using nutrient data where available). The uncertainty on Δ DIC$_{Bio}$, the biological contribution to temporal variations of the surface DIC pool, was calculated by applying standard error propagation to the procedure described in Sect. 4.3.






## 4 Results and discussion

### 4.1 Recent observations (2014-2016)

#### 4.1.1 Surface observations

To characterize the recent state of seawater carbonate chemistry in the study area, we use data obtained in the late summers (August - September) of 2014, 2015 and 2016. Uncertainties for each parameter can be found in the Methods (Sect. 3.3). The mean 2014-2016 surface (< 10 m) practical salinity ($S_P$), TA and DIC values in the Canada Basin (CB), Canadian Arctic Archipelago (CAA) and Baffin Bay (BB) are presented in Table 3. It is important to note that the mean regional values we report for the Canada Basin may be skewed by the higher density of stations located along the Mackenzie Shelf, and that our sample size for Baffin Bay consists of only 6 stations. Practical salinities considerably below 25 were mostly observed near the mouth of the Mackenzie River, with some in the Queen Maud Gulf (QMG). The discrepancy between $S_P$ and TA values observed in the CB/CAA and Baffin Bay clearly illustrates the change in water mass regime west of Lancaster Sound (see Sect. 2), while DIC, which is strongly affected by biological activity, shows a less prominent spatial pattern. In all regions, surface-water $pCO_2$, of which we only consider data acquired over the top 5 meters of the water column in order to render it more indicative of gas exchange potential, was largely undersaturated with respect to the atmosphere, by as much as 150 µatm (Fig. 3). This suggests that the region as a whole acts as a net $CO_2$ sink during the summer, as Geilfus et al. (2018) observed in 2011. Ahmed et al. (2019) came to a similar conclusion based on underway $pCO_2$ measurements carried out in the CAA between 2010 and 2016, but reported consistent temporal fluctuations of summer sea-surface $pCO_2$ and $\Delta pCO_2$ driven by sea-ice processes. The handful of locations exhibiting a positive $\Delta pCO_2$ ($pCO_{2SW} - pCO_{2air}$; outgassing to the atmosphere) are located in the QMG and the transect between Banks Island and Cape Bathurst (Fig. 4), where the episodic upwelling of upper halocline water is well documented (Mucci et al., 2010; Tremblay et al., 2011).

The saturation state of surface waters (<10m) with respect to aragonite ($\Omega_A$, Fig. 5) ranges from 0.82 to 2.03 (mean 1.26), 0.73 to 2.01 (mean 1.22) and 1.47 to 1.69 (mean 1.57) respectively in the CB, CAA and BB. $\Omega_A$ correlates well with $S_P$ (r=0.79; Fig. 6), although the surface samples with the highest salinities ($S_P$ ~32), collected in Baffin Bay, were not the most supersaturated.



Recurrent undersaturated conditions ($\Omega_A < 1$), potentially deleterious to calcifying organisms, are found at the surface in two areas: the central Canada Basin and the Queen Maud Gulf. Such conditions have been well documented in the former (e.g., Robbins et al., 2013), and attributed mostly to the input of sea-ice melt, which lowers $\Omega_A$ by diluting surface waters and promotes $CO_2$

uptake (Yamamoto-Kawai et al., 2011). As of 2016, surface undersaturation with respect to aragonite was limited to the central Canada Basin and did not extend to its bordering continental shelf, where $\Omega_A$ remains near saturation (1-1.25).

In the Queen Maud Gulf, surface $\Omega_A$ values as low as 0.84 and 0.73 ($\pm 0.16$) were observed in

2015 and 2016, respectively (most visible as the two minima in Fig. 6). Most of the surrounding samples were slightly undersaturated ($\Omega_A \sim 0.90 - 1.05$). Both minima were associated with locally higher $pCO_2$ values (~429 and 438 $\pm$ 6 µatm) and practical salinities of 26.54 and 22.17. Two mechanisms, namely the decrease in the calcium ($Ca^{2+}$) ion concentration and alkalinity via dilution by freshwater and the decrease of the $CO_3^{2-}$ concentration concomitant with a decrease in

pH, explain these saturation minima. The former process is self-explanatory and results from the documented increase in freshwater inputs by melting of sea-ice (Yamamoto-Kawai et al., 2009b) and increased river discharge (Déry et al., 2016). We estimate, based on a linear regression of surface $\delta^{18}O$ data against $S_P$, that upwards of 95% of the freshwater at the surface of the QMG in 2015 was of riverine origin (Appendix B). Our dataset does not allow us to directly differentiate

the contributions of air-sea gas exchange and biological activity (respiration) to the high $pCO_2$ observed at these locations. Nonetheless, the depth of these samples (<5 m) implies some degree of equilibration with the atmosphere. Although the diurnal cycle of biological activity may play a role in the development of peaks in $\Omega$, we rule out this mechanism for the case discussed above, as the 0.73 minimum of 2016 was observed in the early afternoon. Although the $\Omega_A$ minima

represent significant undersaturation, the uncertainty on $\Omega_A$ computations (0.08 or 0.16 depending on the parameters used in the calculation; see Methods) blurs the saturation threshold in such a way that $\Omega_A$ values marginally below 1 might in reality represent supersaturated conditions, and vice-versa. It is important to note that, even without the influence of climate change, areas of high riverine discharge naturally harbor lower carbonate mineral saturation states. Thus, undersaturated

conditions in the QMG and elsewhere do not solely result from the documented increase of





freshwater inputs described above. Nonetheless dilution by freshwater affects the degree of this undersaturation as well as its spatial and temporal extent.

Surface waters throughout the study area are supersaturated with respect to calcite, with $\Omega_C$ ranges (mean) of 1.34 to 3.25 (2.04), 1.21 to 3.29 (1.97) and 2.38 to 2.70 (2.52) in the CB, CAA and BB, respectively. Uncertainties on $\Omega_C$ values are on the order of 0.25-0.30, almost twice as large as those of $\Omega_A$, due to the larger uncertainty of the calcite stoichiometric solubility product at 25 °C and $S_P = 35$ (Mucci, 1983).

### 4.1.2 Water column observations

Depth profiles of $pH_T$, $pCO_2$ and $\Omega_A$ grouped by region (equivalent to water-mass regime) are presented in Fig. 7. We divided the broad Canadian Arctic Archipelago region into four sub-regions (see Fig. 2): The Parry Channel, the east-west channel extending from M'Clure Strait to Lancaster Sound, the Amundsen Gulf, the Queen Maud Gulf and the central CAA, which connects the QMG to the Parry Channel. The water mass structures are essentially the same as those described in Sect. 2.

The most prominent feature in profiles of carbonate system parameters in the Canada Basin is the Upper Halocline Layer (UHL), a layer of water originating from the Pacific Ocean with a relatively lower pH due to its high metabolic $CO_2$ content (Shadwick et al., 2011). In 2014-2016, the UHL was characterized by a $pH_T$ minimum of $7.82 \pm 0.03$, a $pCO_2$ maximum of $\sim 652 \pm 6$ µatm and a $\Omega_A$ minimum of $0.75 \pm 0.16$ in the central CB. This $pH_T$ minimum migrates upwards from $\sim 180$ to $\sim 140$ meters as the UHL encounters the continental shelf west of M'Clure Strait but maintains its amplitude. The presence of such an acidified layer exacerbates the vulnerability of the planktonic communities in this area, as, in addition to the aragonite undersaturation found at the surface, $\Omega_A$ drops below one at depths of 100 to 125 meters or even shallower waters in the Canada Basin. As $CO_2$ naturally diffuses or mixes from the UHL to the overlying waters and the combination of gas exchange and freshening continues to generate undersaturated conditions at



the surface, the entire photic zone (where $\Omega_A < 1.5$) may acidify and become undersaturated with

respect to aragonite at a much faster rate than that of other oceans. The shallowest subsurface aragonite saturation horizon we observe in the central Canada Basin was found at ~85 meters in 2014. Our data do not corroborate the interpretation of Wynn et al. (2016) that the upper boundary of the UHL is migrating downwards due to an expansion of the overlying Polar Mixed Layer (PML), at least not during the period of our observations. The $\Omega_A$ crosses the saturation threshold

back to supersaturation between 200 and 250 meters, where Atlantic waters become predominant, as evidenced by a +0.8 °C temperature maximum at ~500 m (in contrast to a temperature minimum of -1.5 °C in the UHL). $\Omega_A$ and $pH_T$ remain, respectively, above 1.3 and 8.05 in this layer, with respective maxima of 1.74 ($\pm$ 0.16) and 8.17 ($\pm$ 0.03) that generally coincide with the temperature maximum mentioned previously. The profiles presented in Fig. 7 do not extend to the bottom of

the basin, but deep waters become undersaturated with respect to aragonite below ~2100 m (data from 2009).

The Amundsen Gulf and the western portion of the Parry Channel (Fig. 7; CAA1) essentially exhibit the same water mass structure and carbonate system chemistry as the Canada Basin, as the

dominant circulation pattern pushes water eastward from the CB to the CAA. Undersaturation with respect to aragonite does not occur at the surface in these areas, owing to higher salinities. Although the amplitudes of the $\Omega_A$, $pH_T$ and $pCO_2$ excursions are slightly smaller than those found in the CB, the UHL is considerably shallower in the western CAA. Consequently, $\Omega_A$ falls below one at depths of 50 to 70 meters, and the upper portion of the water column in those parts of the

CAA might become undersaturated with respect to aragonite even more rapidly than in the CB. As reflected by the blue lines in Fig. 7 (CAA1), the UHL becomes progressively less discernable on depth profiles as it undergoes modification and mixing during its transit from the CB to Lancaster Sound. Atlantic waters are found at the bottom of the water column in the Amundsen Gulf and Parry Channel. The saturation maxima at ~400 m ($\Omega_A$ ~ 1.1 to 1.4) are significantly lower

in these areas than at similar depths in the CB.

The shallow bathymetry of the central CAA restricts the eastern flow originating from the CB to the layer of Pacific water (UHL) and the overlying surface water. $\Omega_A$ is close to 1 at the surface





and reaches a maximum of ~1.45 (± 0.16) at ~30 m, before dipping back below 1 between 50 and

75 m, like in other parts of the CAA. With the exception of one profile that exhibits a strong $pCO_2$

maximum of 685 (± 6.49) µatm as well as $pH_T$ and $\Omega_A$ minima of 7.80 (± 0.03) and 0.76 (±0.16),

respectively, at a depth 125 m, this UHL feature becomes less prominent in the central CAA.

Degradation of settling organic matter (remineralization) might explain this exceptional peak of

greater amplitude than those observed in the Canada Basin at Station 310 in September

2016.Stratification becomes significantly weaker in the shallow waters (20-100 m) of the Queen

Maud Gulf (Fig. 7; CAA2), one of the areas with the strongest tidal mixing in the CAA

(McLaughlin et al., 2004). The residence time of waters in this area might also be relatively high,

due to its geographical isolation from the main channels of the CAA, possibly allowing more

mixing to take place. The combined effects of low salinities from freshwater accumulation (mostly

from river discharge) and the efficient redistribution of $CO_2$ through mixing result in low carbonate

mineral saturation states throughout the water column. The QMG is the only region in our study

area where the entire water column is undersaturated with respect to aragonite, making it an ideal

location to study the effects of such conditions on aragonitic organisms.

East of Barrow Strait, where a sill restricts the eastward flow to the upper 125 meters of the

water column (Bidleman et al., 2007), the water mass regime changes. With the exception of one

profile that captures the Pacific outflow through the western portion of Lancaster Sound at

station 301 (Fig. 7, BB, blue line), this change is clearly visible in profiles of carbonate system

parameters in Lancaster Sound and Baffin Bay, where the upper 500 m of the water column is

supersaturated with respect to aragonite at each of the visited stations. Although surface waters

in Northern Baffin Bay are a mixture of multiple inputs from the CAA and the Arctic Ocean

through Nares Strait, the warm and saline Atlantic water inflow from the Labrador Sea

dominates the water-mass structure in the region and accounts for the high alkalinity of these

waters relative to the CB and CAA (Münchow et al., 2015). In Baffin Bay, waters become

undersaturated with respect to aragonite and calcite at depths of ~600 and ~1400 meters,

respectively. The $pCO_2$ and $pH_T$ increase and decrease proportionally to each other with depth.



### 4.2 Time series

380        Of the 420 stations that make up our dataset, twenty-four were visited at least on two different years and match our comparability criteria for time-series. These criteria are: 1) the stations were sampled within 31 calendar days of each other (this criterion is not ideal since seasonality is highly variable and driven by complex sea-ice processes, including ice break-up) and 2) the stations are located within a 5 km radius. The mean time difference and distance between

comparable stations are 12 calendar days and 1.81 km, respectively. Eight were visited three times, the remainder were visited twice. With the exception of one site in northern Baffin Bay, all recurrently sampled stations are located within or on the outskirts of the CAA. Sixteen of the 24 time-series span 3 years or less. Within all comparable stations, we identified the measurements made at similar depths in the top 100 meters of the water column (Fig. 8). The 2014-2015 and

2014-2016 intervals have the most comparable data points (~30), followed by the 2007-2009 and 2015-2016 intervals. The remaining year combinations have less than 10 comparable data points. The small data pool applicable to time series and the generally short and inconsistent time intervals between comparable data highlight the poor suitability of our dataset to the identification of temporal trends of ocean acidification. The strong interannual variability of carbonate system

parameters in the CAA strengthens this conclusion. Nonetheless, we use data from the seven stations visited over a time interval of at least 5 years (Fig. 9) to identify trends in carbonate system parameters. The resulting depth profiles are shown in Figures 10 and 11.

       To quantify near-surface change, we averaged data from the top 25 meters of the water column. Across this depth interval, between 2007 and 2016, the temperature-normalized $pCO_2$,

DIC and DIC/TA at site LS1 rose by $36 \pm 9$ µatm, $37 \pm 10$ µmol kg$^{-1}$ and $0.008 \pm 0.005$, while $pH_T$ and $\Omega_A$ decreased by $0.042 \pm 0.037$ and $0.12 \pm 0.23$, respectively. Given its uncertainty, the change in $\Omega_A$ is insignificant. The same trend is visible between the same years at station CAA1, at a greater magnitude (+ 78 µatm $pCO_2$, -0.088 $pH_T$ unit, -0.37 $\Omega_A$ unit), except for a DIC decrease of 58 µmol kg$^{-1}$ caused by a decrease in salinity of 1.76. The DIC/TA, which effectively normalizes

DIC against salinity, shows an increase of 0.021, proportional to the change in the other parameters. Data from the nearby station CAA2 also displays a similar trend between 2009 and 2015 (+ 29 µatm $pCO_2$, -0.041 $pH_T$ unit, +0.001 DIC/TA), albeit a strong positive change in



salinity (+1.81) contributed to an increase of 0.06 $\Omega_A$ unit (again, insignificant given its uncertainty of 0.23).


Four stations located on the transect extending from Cape Bathurst to Banks Island complete the seven time-series of at least five years (Fig. 11). Three of those time series (AM1, AM2, AM4) span 2003-2009 or 2004-2009. Thus, given their spatial and temporal proximity, we expect a certain consistency in the trends they exhibit. Surprisingly, salinity increased at each of

the stations, by 1.46, 0.70 and 0.13, respectively. Despite the consistent salinity trend, stations AM2 and AM4 show opposite trends in carbonate system parameters, the former exhibiting a positive change in $pCO_2$ (+ 49 ± 9 µatm) and DIC/TA (+ 0.010 ± 0.005) as well as a negative change in $pH_T$ (-0.089 ± 0.037) and $\Omega_A$ (-0.20 ± 0.23), while the latter displays negative but somewhat smaller changes in $pCO_2$ (- 35 ± 9 µatm) and DIC/TA (- 0.007 ± 0.005) as well as a

positive change in $pH_T$ (+0.051 ± 0.037) and $\Omega_A$ (+0.13 ± 0.23), mostly apparent near the surface (<10 m). The two profiles recorded at site AM2 exhibit a clear difference in stratification; a sharp peak in carbonate system parameters (associated with a temperature maximum) appears at ~25 m in 2003, while the top 50 meters were fairly uniform in terms of $pCO_2$, $pH_T$ and DIC/TA in 2009. The magnitude of change in the carbonate system parameters at stations AM1 (2004-2009) and

AM3 (2009-2016) is below their respective uncertainties. A relatively large change of -1.90 in near-surface salinity is observable at station AM3. The depth profiles at sites AM1 and AM3 (Fig. 11), both of which include measurements from three separate years, highlight the non-linearity of change (i.e., the variation between the earliest and the intermediate year is opposite from the variation between the earliest and latest years). Although this observation is limited to two groups

of profiles, it constitutes a caveat to conclusions drawn from only two points in time.

Below 25 meters depth, variations in atmospheric conditions and biological activity become decreasingly influential on carbonate system parameters, relative to changes in water mass properties resulting from mixing. The largest change in our time series occurs at Station LS1,

where $pCO_2$, DIC and DIC/TA increased by 265 µatm, 95 µmol kg$^{-1}$, and 0.040, and $pH_T$ and $\Omega_A$ decreased by 0.140 and 0.38, respectively, over the 25-100 m depth interval, between 2007 and 2016. This drastic change reflects the varying proportions of Pacific and Atlantic Ocean waters





that reach Lancaster Sound, the latter water mass intruding episodically into the CAA (Prinsenberg and Bennett, 1987). An increase in the abundance (or a change in the properties) of Pacific Ocean

water between 2007 and 2016 is likely the cause of this feature, as a similar trend is visible at station CAA1, located east of Barrow Strait, where the intrusion of Atlantic Ocean water is halted by a 125-m sill. This is supported by the absence of a strong increase in salinity in Lancaster Sound in 2007, which could be interpreted as a pulse of intruding Atlantic Ocean water. This feature is reversed at station CAA2 between 2009 and 2015, highlighting the interannual variability in the

properties of the UHL. Below this layer, water properties are relatively uniform in time at stations CAA1, CAA2, LS1 and AM2, except for $pCO_2$ variations on the order of 20 to 30 µatm. Station AM1 exhibits a strong change in deep-water (300 to 400 m) properties between 2004 and 2008, the latter year displaying considerably less acidic conditions (+ 0.25 $pH_T$ unit), while salinity and temperature profiles are nearly identical for the two years, suggesting vertical carbon export from

the surface as a possible mechanism.

As previously stated, these time series are snapshots in time and cannot be assumed to represent the continuous evolution of the carbonate chemistry in the Canadian Arctic. Nonetheless, even with a small sample size, we can confidently state that the temporal evolution of carbonate

system parameters in the region does not display a systematic trend on sub-decadal timescales. Moreover, most of the significant changes our time series exhibit are associated to variations in the physical oceanography of the region (water-mass distribution and circulation) or surface processes (melting of sea-ice). Given the well-documented rapid melting of the sea-ice cover in the region (e.g., Tivy et al., 2011), we did not expect to observe increases in summer surface

salinity over time intervals of 5 to 9 years. Our time series, therefore, offer proof of the strong interannual variability of this highly dynamic system. Discerning the ocean acidification signal amid the various physical and biological sources of change would require continuous time series over a longer period of time. We estimated this period to 23 to 35 years for pH, 25 to 37 years for pCO2, 31 to 46 for ΩA and 118 to 177 years for DIC using calculations of Time of Emergence, or

the time required for the effects of a process to emerge from the natural variability of a system (see Appendix A). Time of emergence calculations are usually performed with large, continuous datasets from climate models; we therefore do not consider these results to be statistically correct.



Nonetheless, this exercise shows the relative variabilities of the carbonate system parameters and highlights the particularly strong variability of DIC, which is the object of the next section. Its

results also imply that without accurate measurements of the effects of biological activity and sea-ice processes (both major drivers of natural variability), direct detection of the ocean acidification signal will require at least 20-25 years of observations. Gathering data in the CAA in the next few years is therefore critical, as regular ship-based observational campaigns in the region started in the early 2000's (Giesbrecht et al., 2014).


### 4.3 The biological contribution to interannual DIC variations: $\Delta DIC_{Bio}$

We define $\Delta DIC_{Bio}$ as the change in the contribution of in-situ biological activity (photosynthesis and respiration) to the DIC of a parcel of water over a given period of time. $\Delta DIC_{Bio}$ is calculated for each depth at recurrently visited stations according to:

$$\Delta DIC_{BIO} = DIC_{Observed} - DIC_{Reference}$$

where $DIC_{Reference}$ is computed in CO2SYS using the temperature-normalized seawater $pCO_2$ calculated at a reference time (from measured TA and $pH_T$) and the TA measured at the time of interest (the year for which $\Delta DIC_{Bio}$ is reported). The change in global mean atmospheric $CO_2$ concentrations between the reference year and the year of interest is added to the $pCO_2$ in order to

account for gas exchange (data from Dlugokencky and Tans, NOAA/ESRL). This approximation rests on the assumption that the yearly increase in surface water $pCO_2$ follows that of the atmosphere (given stable biological production), as observations from global monitoring stations demonstrate (e.g., González-Dávila et al., 2010), although the validity of this claim is weakened on short spatial and temporal (sub-decadal) scales (Fay et al., 2013, Wanninkhof et al., 2013). This

also restricts our calculations to the upper portion of the water column (25 m) that is in direct contact with the atmosphere. Under the additional assumptions that DIC is only affected by gas exchange, biological activity and mixing, and that TA is not significantly affected by biological activity (Zeebe and Wolf-Gladrow, 2001), $DIC_{Reference}$ represents the DIC of a parcel of water if its *in-situ* biological component remained unchanged relative to a reference year (i.e., identical

contribution, negative or positive, from the balance between photosynthesis and respiration). Because the reference $pCO_2$ is calculated in part from TA, changes in water masses should not





affect the results of this analysis, given the salinity range of the data subset used in the calculation of $\Delta DIC_{Bio}$ ($25.6 < S_P < 33.7$). Thus, $\Delta DIC_{Bio}$ can provide insights into the interannual variability of biological activity in the Canadian Arctic, without direct measurements of parameters such as

chlorophyll or biomass.

Figure 12 shows $\Delta DIC_{Bio}$, averaged over the top 25 meters of the water column, at the 18 stations where comparable data were available, all located in the Amundsen Gulf and CAA. The magnitude of the calculated $\Delta DIC_{Bio}$ is beyond its uncertainty, which varies from 6.4 to 14.3 µmol

kg$^{-1}$ (mean of $\pm$ 8.6 µmol kg$^{-1}$), at 11 locations out of 18. All significant results from the month of October (2003-2009 and 2007-2009) in the Amundsen Gulf show a positive mean $\Delta DIC_{Bio}$ (a decrease in biological DIC uptake and/or an increase in DIC production) of up to $20.6 \pm 10.5$ µmol kg$^{-1}$. Conversely, the $\Delta DIC_{Bio}$ calculated for the month of August (2004-2009, 2009-2016 and 2014-2015) is generally negative and of a similar magnitude. Data for September are variable in

direction and show the greatest change over consecutive years: relative to 2014, in the Amundsen Gulf, the mean $\Delta DIC_{Bio}$ fluctuated in 2015 and 2016 by $+28.0 \pm 11.0$ to $-11.2 \pm 8.0$ µmol kg$^{-1}$ over the 0-25 m depth interval.

It is unlikely that the consistent directions of change we observe for the months of August

and October represent actual trends, given the small data pool and inconsistent reference states used to make those observations. Nevertheless, we can use this analysis to constrain the contribution of fluctuations in biological activity to the interannual variability of the DIC. In the top 25 meters of the water column, the maximum amplitude of $\Delta DIC_{Bio}$ is $45.9 \pm 11.5$ µmol kg$^{-1}$, observed in the Amundsen Gulf between 2014 and 2015. This maximum is not visible on Fig. 14,

which only shows mean values. The Pearson correlation coefficient (r) between $\Delta DIC_{Bio}$ and $\Delta DIC$ (the change in DIC over the same time interval) is 0.52 and the associated coefficient of determination ($R^2$) is 0.27, suggesting that 27% of the interannual variability in surface DIC in the Canadian Arctic Archipelago is a direct result of the variability in biological activity. This has important implications with respect to the interpretation of DIC time series in the region and

explains, in part, the long time of emergence of DIC as a proxy for acidification, compared to other carbonate system parameters, as hypothesized in section 4.2.



The extremely weak (r = 0.08) correlation between $\Delta DIC_{Bio}$ and the time interval over which it applies provides additional evidence of the absence of a trend in the balance between

photosynthesis and respiration in the surface waters of the Canadian Arctic. The variability in this balance is driven by many interconnected, often localized processes. For instance, short-lived episodes of upwelling of halocline waters not only directly change the chemical properties at the surface, but also provide nutrients that stimulate biological activity (Tremblay et al., 2011). Primary production in the Arctic is also closely linked to the seasonal cycle of sea-ice (e.g., Arrigo

et al., 2008). Beyond its natural fluctuations, ongoing disruptions to this cycle and other physical properties (stratification, temperature, etc.) forced by climate change are known to affect phytoplankton communities and their productivity (Ardyna et al., 2014; Blais et al., 2017), possibly increasing their imprint on the variability of DIC and other carbonate system parameters. As previously mentioned, variations in water mass composition cannot directly explain variations

in $\Delta DIC_{Bio}$. Nonetheless, mixing is likely accompanied by changing nutrient concentrations, which influence the balance of photosynthesis and respiration (Tremblay et al., 2015).

## 5 Conclusions

Field observations of carbonate system parameters made between 2014 and 2016 in the

Canadian Arctic reveal that surface waters of the region serve as a net $CO_2$ sink in the summer and are generally close to saturation with respect to aragonite ($1 < \Omega_A < 1.5$). Surface undersaturation ($\Omega_A < 1.0$) is found predominantly in the central Canada Basin, as documented in previous years (Yamamoto-Kawai et al., 2011; Robbins et al., 2013), and in the freshwater-influenced Queen Maud Gulf, the only area of the Canadian Arctic where the entire water column is undersaturated

with respect to aragonite. The $CO_2$ – rich Upper Halocline Layer is the locus of shallow aragonite undersaturation horizon at depths of 85 to 125 m in the Canada Basin and 50 to 70 m in the Canadian Arctic Archipelago, further fostering undersaturation near the surface and potentially threatening marine calcifying organisms living in this portion of the water column.



Time series of carbonate system parameters, although relatively short (<10 years) and incomplete, illustrate the strong interannual variability of the region, due in part to complex circulation patterns and varied water mass assemblages. Our estimates of $\Delta DIC_{BIO}$, the change in the contribution of biological activity to DIC, suggest that variations in biological activity (the balance between photosynthesis and respiration) account for approximately a third of the

interannual variability of DIC measurements. Additional work must also be carried out to extend this estimate to other carbonate system parameters (pH, $pCO_2$, $\Omega$). In order to test the validity of the $\Delta DIC_{BIO}$ concept and its underlying assumptions, our results should be compared to direct measurements of biological productivity (e.g. biomass, chlorophyll-a) during the same time period. Without the latter, the quantification of the progression of ocean acidification in the surface waters

of the Canadian Arctic will require longer and more continuous time series, the length of which can be estimated using the concept of Time of Emergence. Future work on ocean acidification in this region should focus on obtaining continuous time series of carbonate system parameters, especially in areas where surface waters might soon become undersaturated with respect to aragonite, as well as bridging the gap between observations of carbonate mineral saturation and

markers of ecosystem heath.







## Appendix A

**Time of Emergence of ocean acidification signals**

The time of emergence (ToE) of a process affecting a natural system is the time required for the measurable effects of this process to emerge from the natural variability of the system. The concept is predominantly applied in global climate change modeling studies, for which the results are either "years of emergence" based on a pre-industrial steady-state (e.g., Friedrich et al., 2012) or time intervals over which observations must be made in order to distinguish an anthropogenic signal from natural variability. Few of these studies have used observations (e.g., Sutton et al. 2016), and, to our knowledge, none of them have focused specifically on the Arctic.

We define the time of emergence according to the following equation:

(A1)
$$ToE = \frac{C \times N}{S}$$

in which N is the natural variability, defined (after Hawkins and Sutton, 2012) as the standard deviation of the annual means of one of four parameters (DIC, T-normalized $pCO_2$, $\Omega_A$, $pH_T$), across 7 years from 2003 to 2016. $S$ is the slope (in units per year) of the theoretical acidification curve, constructed by calculating the evolution of a given parameter in CO2SYS, at a constant alkalinity (2000 $\mu$mol kg$^{-1}$), practical salinity (30) and temperature (0°C), and assuming that the surface ocean $pCO_2$ increases at the same mean rate as the atmospheric $pCO_2$ between 2003 and 2016 (data from Dlugokencky and Tans, NOAA/ESRL). Our computed rates of acidification (Table A1) are in general agreement with the values reported by Bates et al. (2014). C is a constant that sets the threshold of emergence at either 2 or 3 standard deviations (N), i.e., when the acidification signal becomes significant beyond natural variability as it emerges from 95% or 99.7% of the observed annual mean values, assuming the data (or the naturally occurring values of the parameter they represent) are normally distributed. Only data collected from June to October, inclusively, are used to minimize the effect of seasonal variability.

605



Although the assumption of relative equilibration with atmospheric $pCO_2$ might be less applicable at these depths, we applied the same technique using data from depths of 100 m and 300 m in order to estimate the time of emergence below the surface, where the interannual variability should be relatively small. Significant acidification at these depths might be due in large part to advection (e.g., Luo et al., 2016) rather than gas exchange.

The results of this analysis are presented in Table A2. At the surface, $pH_T$ and $pCO_2$ have very similar times of emergence of 23-35 years and 25-37 years for C values of 2 and 3, respectively. The $\Omega_A$ signal emerges after a slightly longer time, 31-46 years. The ToE of DIC is considerably longer than that of the three other parameters, at 118-177 years. The discrepancy between DIC and the other parameters is largely due to chemical considerations, as the high Revelle factors (15-20) in the region imply that the change in $pCO_2$ caused by the uptake of atmospheric $CO_2$ will be 15 to 20 times larger than that of DIC, relative to their initial concentrations.

Our results show a slight increase of ToE values from the surface to the 90-110 m depth interval (except for $\Omega_A$ that remains identical), followed by a decrease below surface values at the 290-310 m depth interval (except for DIC). We attribute the relatively strong variability of the ToE in the 90-110 m depth interval to a periodical vertical migration of the upper boundary of the metabolic $CO_2$-rich (low $pH_T$, $\Omega_A$, high $pCO_2$, DIC) Upper Halocline Layer (UHL) that is found at depths of ~100-200 m throughout most of the study area. The results are also likely affected by the decreasing number of data with depth (Table A3), that could lead to less accurate annual means and explain the increase of the DIC ToE from the surface to the 290-310 m interval.

Despite their similarity, our calculated times of emergence are consistently longer than those reported in modelling studies (Keller et al., 2014; Rodgers et al., 2015) This is consistent with the fact that coastal waters, that comprise a large portion of our dataset, exhibit a much higher variability in pH (and other carbonate system parameters) than open oceans (Duarte et al. 2013). Furthermore, direct observations are likely to integrate variability on temporal and spatial scales that are too small to be resolved by models. It is also important to note that distinct measurement

635 techniques and their associated uncertainties create an analytical bias between different parameters, a bias that is not present in the same form in modelling studies. The relative uncertainties of *in-situ* $pH_T$ (0.3%) measurements as well as computed $pCO_2$ (4.5%) and $\Omega_A$ (6.4%) values are strongly correlated with their increasing times of emergence in the 0-25 m depth interval (r=0.88). Although this correlation becomes significantly weaker (r=0.49) when DIC and its

640 relative uncertainty (0.4%) are included in the analysis, it suggests the presence of an analytical bias in our ToE estimates. Nonetheless, the similarity between the ToE of $pH_T$ and $pCO_2$ as well as the large ToE of DIC are consistent with the findings of Keller et al. (2014).

 The validity of these conclusions depends on a methodology that differs considerably from its

645 modelling equivalent, even if the results from both approaches are consistent with each other. In addition to the instrumental bias mentioned previously, our observations are subject to a sampling bias, since we only use data gathered in the summer months. Consequently, the natural variability used in our TOE calculations does not encompass the entire annual cycle. Nevertheless, because we define the natural variability of the system in terms of interannual rather than seasonal changes,

650 the former should not change, assuming the amplitude of the seasonal cycle is constant through time. The other form of sampling bias possibly affecting our results is spatial, as cruise tracks and durations varied every summer.

**Appendix B**

655 **Estimation of the freshwater sources in the Queen Maud Gulf**

 In order to estimate the relative fractions of sea-ice melt and meteoric water (mostly river water) in the Queen Maud Gulf, we use $\delta^{18}O$ and practical salinity ($S_P$) data collected in the area in 2015 to perform a linear regression analysis. The methods of analysis of the oxygen isotopes, reported on the $\delta^{18}O$ notation (V-SMOW), are described in detail in Lansard et al. (2012) for the samples

660 collected during the CASES expedition (2003-2004) and in Mucci et al. (2018) for the remainder of the samples.

Using the intercept of the trendline equation ($\delta^{18}O = 0.5282S_P - 18.552$) to extrapolate $\delta^{18}O$ to $S_P$ = 0, we find the mean $\delta^{18}O$ of the freshwater found in the Queen Maud Gulf in 2015 to be -18.55 ‰. Based on the $\delta^{18}O$ values of meteoric water (-18.9 ± 0.1 ‰) and sea-ice melt (-2.0 ± 0.5 ‰)

used by Lansard et al. (2012), the fractions of meteoric (river) water and sea-ice melt would be, respectively, 98% and 2%. A potential source of error affecting this estimate is the use of the $\delta^{18}O$ of Mackenzie River water as the riverine end-member, which might differ significantly from the oxygen isotope signature of the rivers discharging in the Queen Maud Gulf.

**Data availability**

The raw data collected as a part of the ArcticNet program, on which most of the observations presented in this paper are based can be accessed through the Polar Data Catalogue (Mucci, 2017). Complementary datasets, some of which are part of larger databases, are also available on various online repositories (François et al., 2012; Chierici et al., 2013; Giesbrecht et al., 2014;

Papakyriakou et al., 2017).

**Author contribution**

A.M. and A. B.-L. conceived the project. A.M. and H.T. acquired much of the data prior to 2016. A. B.-L. carried out the data analysis and wrote the first draft of the paper whereas A.M. and H. T.

provided editorial and scientific recommendations. H.T. provided results of alkalinity and dissolved inorganic carbon analyses and scientific recommendations.

**Competing interests**

The authors declare that they have no conflict of interest.




## Acknowledgements

We would like to thank the Captains and crew of the CCGS Amundsen without whom,
over the years, this project would not have been possible. This project was funded through
ArcticNet, itself funded by the Natural Sciences and Engineering Research Council of Canada
(NSERC), the Canadian Foundation for Innovation (CFI) as well as the IPY-NSERC, CCAR-
NSERC and French MALINA programs. Additional funding came from a Regroupement
Stratégique grant from the Fonds Québécois de Recherche Nature et Technologies (FQRNT) to
GEOTOP as well as NSERC Discovery and Marine Environmental Observation, Prediction and
Response Network (MEOPAR; Canadian Ocean Acidification Research partnership) grants to
A.M and H.T. We would like to thank Dr. Jean-Francois Hélie at GEOTOP-UQAM for carrying
out the $\delta^{18}O_{(H2O)}$ analyses as well as Constance Guignard for cruise preparation and support in the
laboratory and at sea. Finally, A. B.-L. wishes to thank MEOPAR and the Department of Earth
and Planetary Sciences at McGill for financial support in the form of stipends, scholarships (e.g.,
Mountjoy Scholarship) and assistantships. Fig. 1 in this study was created with the Ocean Data
View Software (Schlitzer, 2016).

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



**Table 1:** Research cruises carried out by the CCGS Amundsen from which the

dataset was constructed.

| Cruise | Timescale |
|---|---|
| CASES | September 2003 - August 2004 |
| CFL | September 2007 - July 2008 |
| MALINA | July - August 2009 |
| GEOTRACES | August - September 2009 |
| ARCTICNET Leg 4a | October - November 2009 |
| ARCTICNET | August - September 2013 |
| ARCTICNET | August - September 2014 |
| GEOTRACES / ARCTICNET | July – September 2015 |
| ARCTICNET | August - October 2016 |






**Table 2**: Mean uncertainties of parameters computed in CO2SYS, their standard deviations and their relative weight with respect to the mean value of each parameter, for different carbonate parameter input pairs. * The $\Delta DIC_{Bio}$ uncertainty depends on whether $DIC_{Observed}$ was measured or computed.


| Computed parameter | Mean uncertainty | Uncertainty standard deviation | % of mean value |
|---|---|---|---|
| $pH_T$ (i-s) | 0.026 | 0.002 | 0.3 |
| DIC ($\mu mol\ kg^{-1}$) | 6.8 | 0.7 | 0.4 |
| Input Parameters: TA/$pH_T$ (TA/DIC) | | | |
| $pCO_2$ ($\mu atm$) | 6.5 (25.0) | 1.8 (6.7) | 4.5 (17.5) |
| $\Omega_C$ | 0.27 (0.13) | 0.04 (0.04) | 6.5 (3.2) |
| $\Omega_A$ | 0.16 (0.08) | 0.02 (0.02) | 6.4 (3.1) |
| $\Delta DIC_{Bio}$ ($\mu mol\ kg^{-1}$) | 7.8-8.6* | 1.3 | 69-77* |





**Table 3**: Mean surface-water (<10 m) practical salinity ($S_P$), total alkalinity (TA), dissolved organic carbon (DIC) and associated ranges in the three sub-regions of the study area in 2014, 2015 and 2016.

| | $S_P$ | | TA (µmol kg$^{-1}$) | | DIC (µmol kg$^{-1}$) | |
|---|---|---|---|---|---|---|
| | mean | range | mean | range | mean | range |
| **Canada Basin** | 27.52 | 19.11 - 30.82 | 2029 | 1820 - 2231 | 1929 | 1761 - 2066 |
| **CAA** | 27.57 | 22.17 - 31.40 | 1967 | 1604 - 2194 | 1868 | 1553 - 2061 |
| **Baffin Bay** | 30.96 | 29.69 - 32.10 | 2138 | 2062 - 2209 | 1999 | 1920 - 2058 |







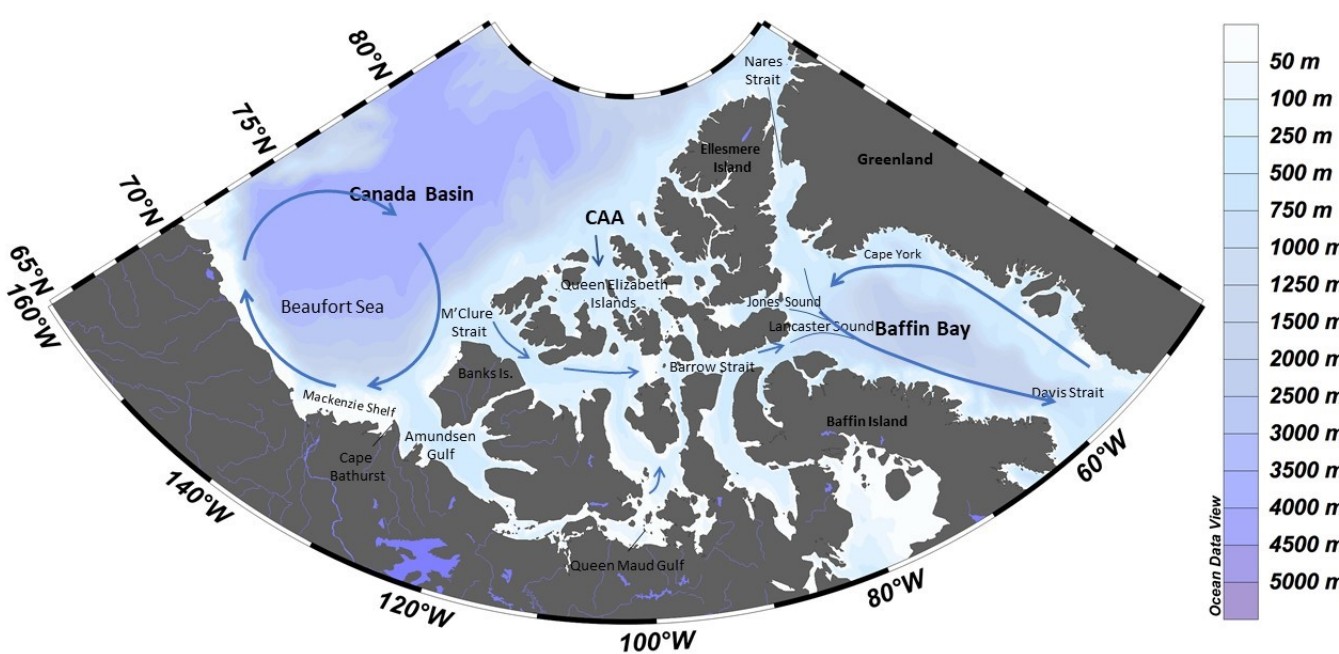

**Figure 1**: Map of the study area with dominant surface circulation flow paths. After McLaughlin et al. (2004) and Proshutinsky et al. (2009). CAA: Canadian Arctic Archipelago. Created using Ocean Data View (Schlizter, 2016)







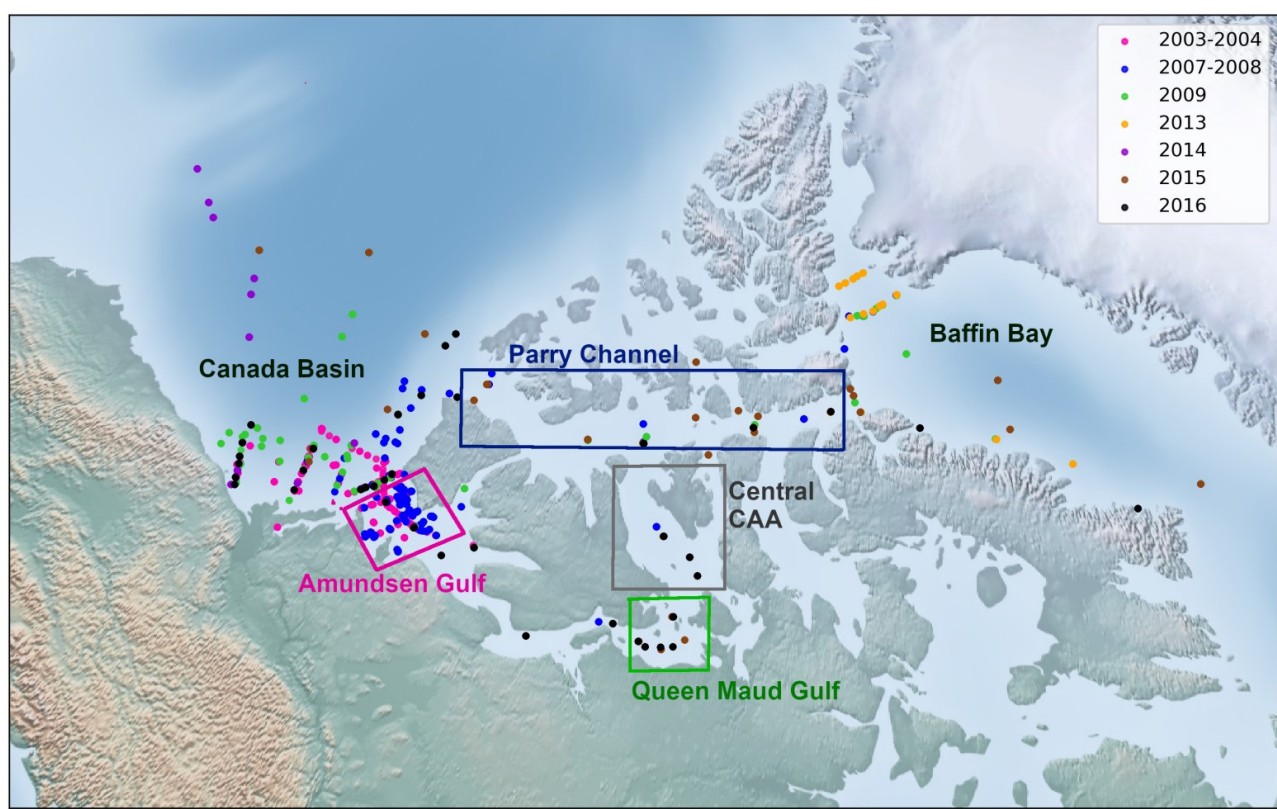

**Figure 2**: Geographical location of the oceanographic stations covered by the dataset, color-coded according to the year of sampling, with the approximate boundaries of the main areas mentioned in the text.




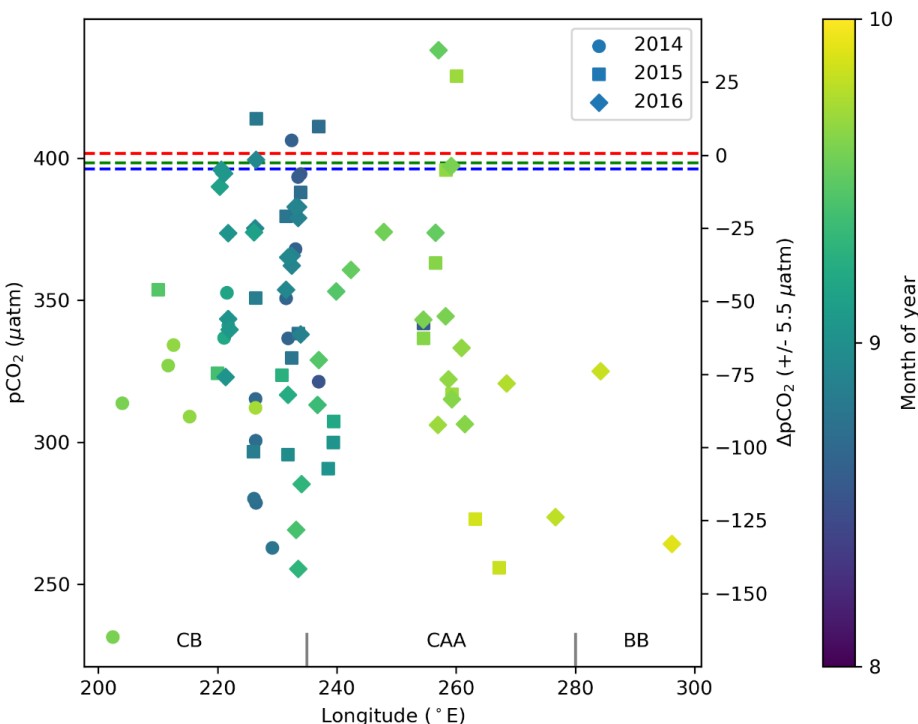

**Figure 3**: Surface-water (<5m) $pCO_2$ and $\Delta pCO_2$ ($pCO_{2SW} - pCO_{2air}$) from 2014, 2015 and 2016 in the Canada Basin (CB), Canadian Arctic Archipelago (CAA) and Baffin Bay (BB). The blue, green and red dashed lines represent the average of the global mean monthly atmospheric $pCO_2$ values for August and September 2014, 2015 and 2016, respectively (data from Dlugokencky and Tans, NOAA/ESRL). The $\Delta pCO_2$ axis shifts in accordance with the atmospheric value used for each year, such that points may be offset from their true position by up to 5.5 µatm.





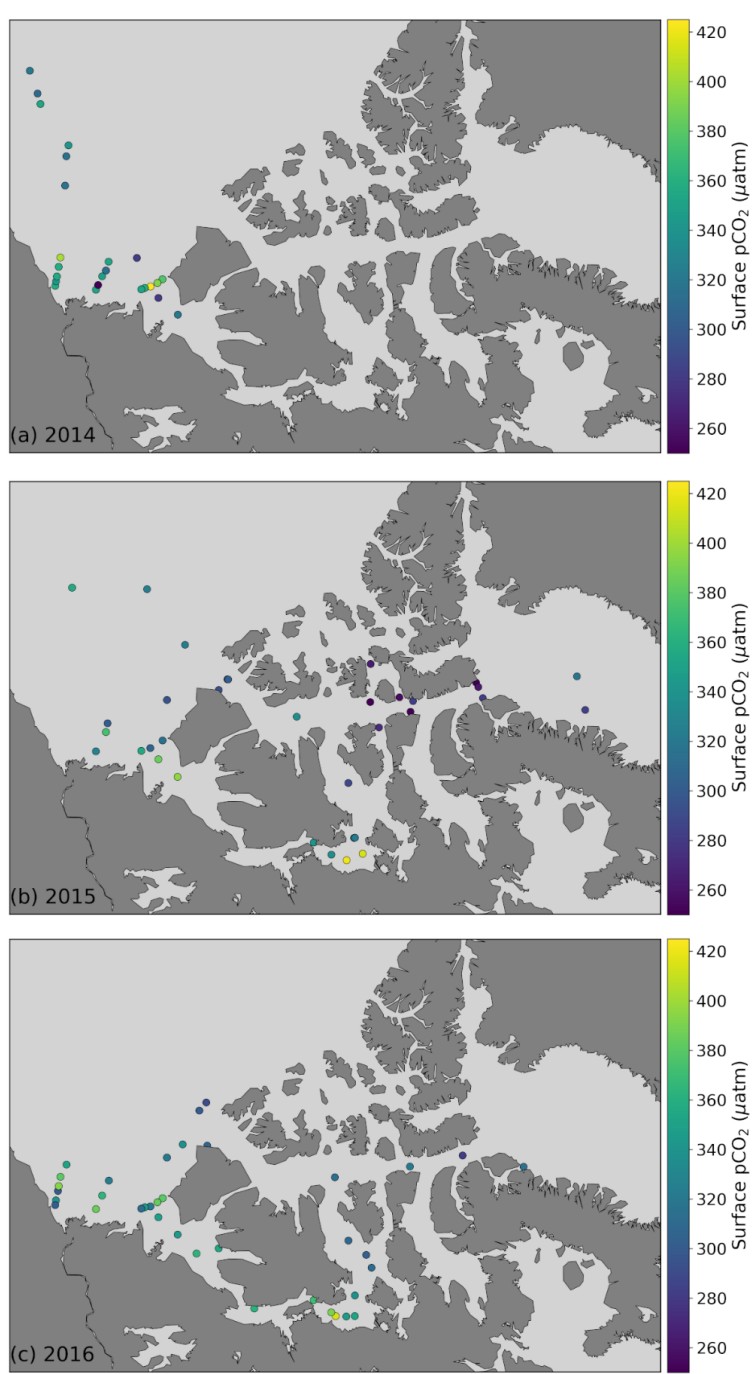

**Figure 4**: Surface-water (<5m) pCO₂ in 2014 (a), 2015 (b) and 2016 (c).

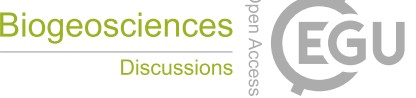

**Figure 5**: Surface-water (< 10m) $\Omega_A$ (left) and $\Omega_C$ (right) in 2014 (top), 2015 (middle) and 2016 (bottom). Note the different color scales for $\Omega_A$ and $\Omega_C$, for which the diverging value for the former is 1, 2 for the latter.




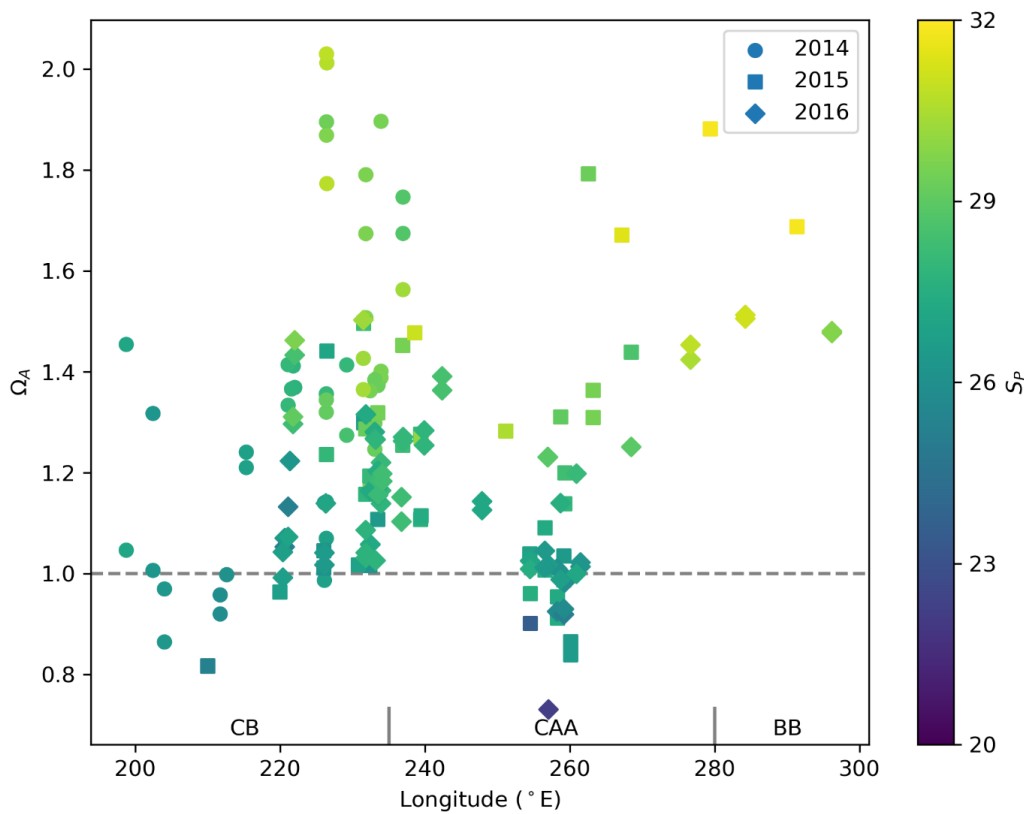

**Figure 6**: Surface-water (< 10m) $\Omega_A$ and associated practical salinities ($S_P$, color scale) in 2014,

2015 and 2016 in the Canada Basin (CB), Canadian Arctic Archipelago (CAA) and Baffin Bay

(BB). The horizontal line at $\Omega_A = 1$ represents saturation with respect to aragonite.










**Figure 7**: Depth profiles of $pH_T$ (left), $pCO_2$ (middle) and $\Omega_A$ (right) at representative stations of the Canada Basin (CB; top), Baffin Bay (BB; bottom) and the Canadian Arctic Archipelago (CAA). The latter is divided into 4 sub-regions (shown on Fig. 2): Parry Channel, Amundsen Gulf, Central Archipelago and Queen Maud Gulf (QMG).




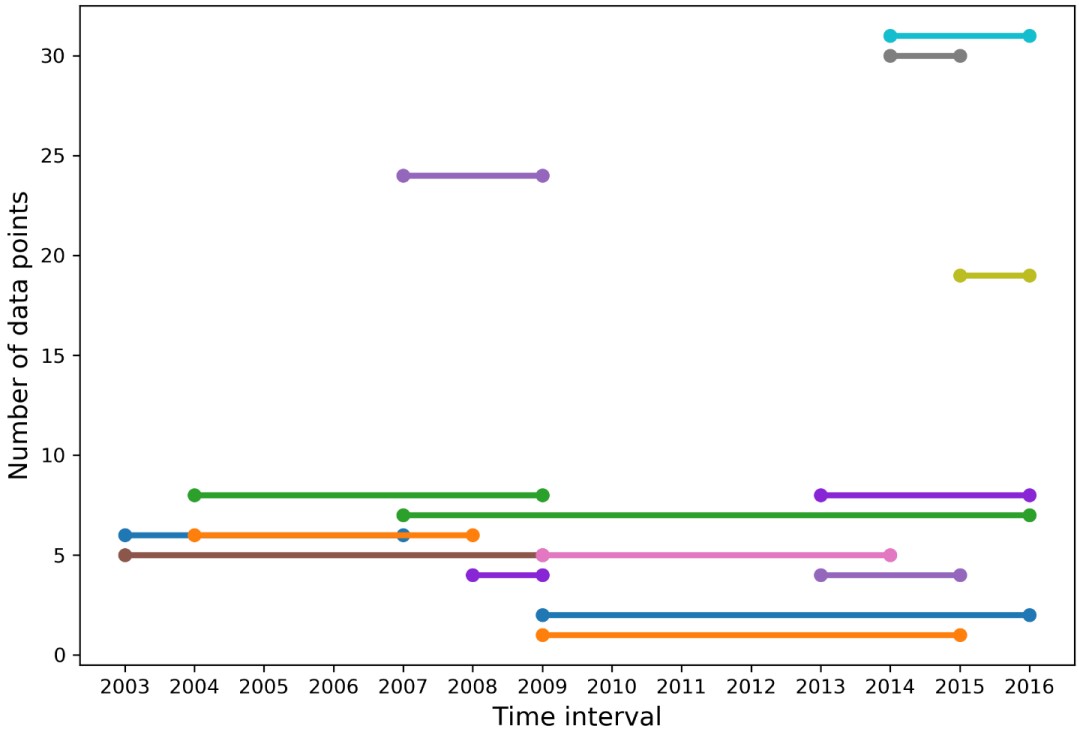

**Figure 8:** Number of comparable data points in the top 100m of the water column, obtained from measurements made at the same station and within set depth intervals (of 3.5 meters at the surface, increasing progressively to 15 m to a depth of 100 m), between each year of the dataset.




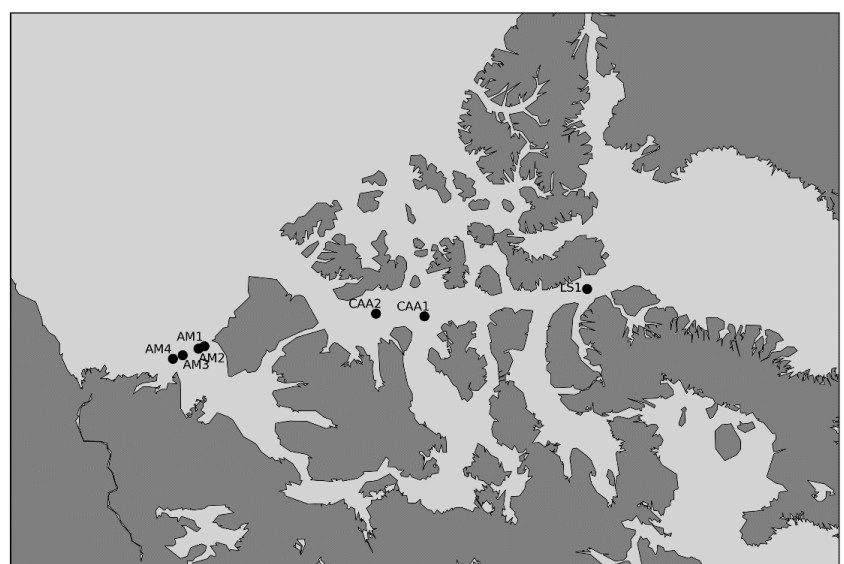

**Figure 9:** Location of the stations visited over an interval of 5 years or more. Stations are designated by area-related acronyms (LS: Lancaster Sound, CAA: Canadian Arctic Archipelago, AM: Amundsen Gulf), that are not the station identifiers used during the expeditions.





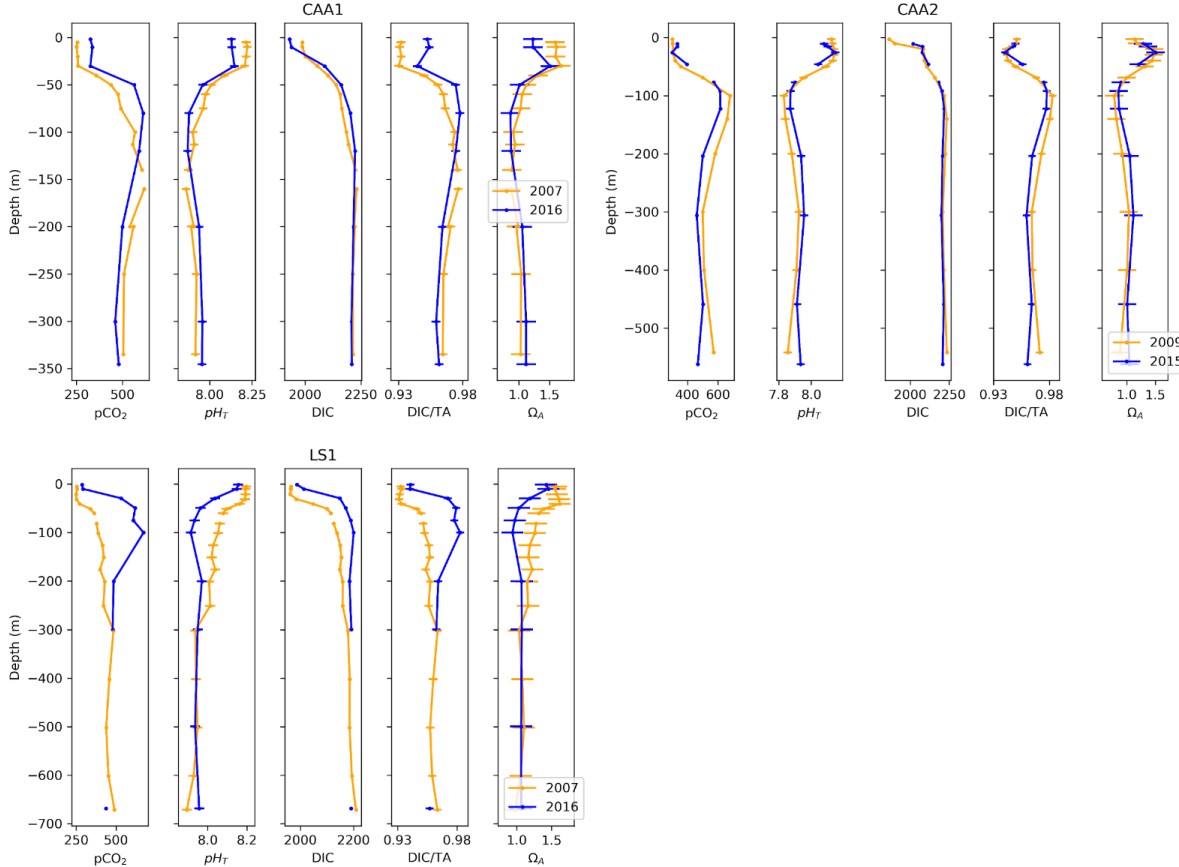

**Figure 10:** Depth profiles of pCO₂, pH$_T$, DIC, DIC/TA and $\Omega_A$ across time at one location in Lancaster

Sound (LS1) and two locations in the Canadian Arctic Archipelago (CAA1, CAA2).








**Figure 11:** Depth profiles of pCO$_2$, pH$_T$, DIC, DIC/TA and $\Omega_A$ and uncertainties across time at four locations at the western limit of the Amundsen Gulf (AM1,2,3,4).





**Figure 12**: Mean $\Delta DIC_{Bio}$ (top 25 meters of the water column) at stations of the Amundsen Gulf (top) and Canadian Arctic Archipelago (bottom). The central dashed line (0) is the reference state, in relation to which $\Delta DIC_{Bio}$ is expressed (i.e. the point at the right extremity of a line indicates the $\Delta DIC_{Bio}$ relative to the year at which this line intersects the zero dashed line). The grey shaded area represents the range of uncertainty.





**Table A1**: Values of natural variability (N; the standard deviation of the annual means of each

parameter) and theoretical acidification rate (S; the slope of the theoretical acidification curve)

used in the time of emergence calculations.

| Parameter | N | S |
|---|---|---|
| pH (i-s) | 0.025 | $-0.002$ yr$^{-1}$ |
| pCO$_2$ | 26.65 µatm | 2.17 µatm yr$^{-1}$ |
| Ω$_A$ | 0.07 | $-0.004$ yr$^{-1}$ |
| DIC | 36.37 µmol kg$^{-1}$ | 0.61 µmol kg$^{-1}$ yr$^{-1}$ |








**Table A2**: Calculated times of emergence of carbonate system parameters at various depths, rounded to the closest year. The C value represents the threshold used (2 or 3 standard deviations of annual means).


| Parameter | | Time of Emergence (Years) | | |
|---|---|---|---|---|
| | | 0 – 15 m | 90 – 110 m | 290 – 310 m |
| pH (i-s) | C = 2 | 23 | 26 | 21 |
| | C = 3 | 35 | 40 | 31 |
| $pCO_2$ | C = 2 | 25 | 36 | 20 |
| | C = 3 | 37 | 55 | 29 |
| $\Omega_A$ | C = 2 | 31 | 31 | 20 |
| | C = 3 | 46 | 46 | 29 |
| DIC | C = 2 | 118 | 156 | 138 |
| | C = 3 | 177 | 234 | 207 |





**Table A3**: Number of data points included in the annual means used in the time of emergence

calculations.

| Year | Data Count | | |
|---|---|---|---|
| | 0 – 15 m | 90 - 110 m | 290 – 310 m |
| **2003** | 166 | 30 | 15 |
| **2004** | 207 | 47 | 22 |
| **2007** | 65 | 24 | 10 |
| **2008** | 146 | 66 | 31 |
| **2009** | 103 | 17 | 10 |
| **2014** | 63 | 20 | 16 |
| **2015** | 24 | 10 | 5 |
