# Peer review of "The recent state and variability of the carbonate system of the Canadian Arctic Archipelago and adjacent basins in the context of ocean acidification"

_Biogeosciences, 2020_

## Referee Comment (RC1) · Leif Anderson (Referee) · 9 Mar 2020

This manuscript includes a very thorough data assessment of the carbon system parameters in Canadian Arctic waters and clearly shows the large variability both in time and space. It confirmed already published conclusions that much of Arctic shallow waters are undersaturated with respect to calcium carbonate, especially aragonite. The basis for this fact is both lowering of salinity by river runoff and sea ice melt in the summer, as well as degradation of organic matter, both in the runoff and in the shelf seas. A valuable contribution is evaluation of "time of emergence" that shows the need for several decades of data in order to separate variability from trends.

[Figure]

Another novel approach is to compute the biological contribution to DIC time variations in the top 25 m, i.e. how much of the ocean acidification signal that is of biological activity. They find this to be about one quarter; a number that I argue has to be taken with great care. The reason is that the DIC-reference computation assumes a constant difference in air-sea pCO2 with time. They mention this but do not relate this assumption to the sea ice coverage before and at the time of measurement. In this region of very temporal variation in sea ice coverage this could have a substantial impact. In the manuscript the decrease in sea ice coverage in the Canadian Basin is mentioned as a cause of elevated uptake of atmospheric CO2, so the authors are aware of this aspect. One could use the satellite images to assess potential changes in the sea ice cover of the region at the times of interest, if these data have enough special and temporal coverage. Considering the fact of changing sea ice coverage I am not sure that the statement on line 504 that the calculated delta DIC-bio is beyond the uncertainty. Further, the computation of delta DIC-bio, as described from line 480, is described in a complicated way. I had to read this section several times before understanding what had been done. My understanding is that DIC-ref is computed from TA and pH-t at the starting date, and then corrected for the increase in DIC by the increasing atmospheric pCO2, assuming a constant air-sea delta value. This could be described in a strait forward way.

Generally this manuscript is well written, but there are several more elements that need more care.

In line 48 it reads "of these phenomena". What do these refer to?

Line 109 reads "and expanding undersaturation". Of what? I assume omega, but as CO2 is also discussed in this section one has to be explicit.

Line 165. m-Cresol purple was used for pH determination. Specify if it was purified version or not as this have significant impact on the accuracy, even when calibration against TRIS buffer. Information on the magnitude of correction of the measured value

after calibration should be added.

In line 228 DIC-bio is mentioned before it has been defined. Hence move it to the section where it is defined.

Sentence starting on line 242 does not specify depth range. I assume it refers to the surface water, and if so it should be specified.

Line 258. I don't see the meaning in giving mean values of here as the range is so large and the mean is very much depending on the number of samples included in the different sub-regions. This also refers to the data on line 295.

Sentence starting on line 282 is confusing. What do you rule out? That air-sea exchange has not increased the omega value, or that primary production has not? Specify.

When it refers to Fig. 7 in the text it would be useful to also give the letters of the individual sub-figures.

Line 311. I assume that the pCO2 and pH-t values are at in situ temperature, but it would be nice to have this specified.

The statements starting with the sentence on line 355 needs some more explanations. As salinity is not presented (more than as averages in the table) the reader cannot judge the statement of weaker stratification. So how does this weakening agrees with the statement on line 359 of low salinities from freshwater accumulation followed by efficient mixing? I would assume that addition of freshwater to the surface would strengthen stratification and thus hampering mixing. Please clarify.

Line 362. It states that the entire water column is undersaturated, but in the figure not all are. Same goes for line 549 in conclusions. Change.

Sentence starting on line 439. Such conditions would gain on being compared with the dominating atmospheric pressure fields at the time (and before) the observations.

Suggestion ending the paragraph on line 449. I would challenge this explanation as the 2004 pCO2 data seems quite close to atmospheric levels, while the 2008 data are substantially below. Also the data at 140 m (?) show a difference between the years that is similar. The two options to get undersaturated CO2 is cooling or primary production. How do you get this signature down to 300-400 m depths? I can only see that there must be two different water masses present (if the data are correct that I do not doubt).

Line 464. Make 2 and A subscript in pCO2 and omega A

Line 608. The statement that the interannual variability should be relatively small at these depths is not necessarily true. In this dynamic region the contribution of different water masses might vary significantly also at these depths.

Finally I was surprised that there is no reference to the below article that is highly relevant to this contribution. Azetsu-Scott, K., A. Clarke, K. Falkner, J. Hamilton, E.P. Jones, C. Lee, B. Petrie, S. Prinsenberg, M. Starr and P. Yeats, 2010. Calcium carbonate saturation states in the waters of the Canadian Arctic Archipelago and the Labrador Sea. Journal of Geophysical Research, 115:C11021, doi:10.1029/2009JC005917.

---

## Referee Comment (RC2) · Anonymous Referee #2 · 24 Mar 2020

The Arctic Ocean is particularly vulnerable to acidification due to its relatively low buffering capacity and, thus is considered as a bellwether to study global ocean acidification. The manuscript "The recent state and variability of the carbonate system of the Canadian Arctic in the context of ocean acidification" written by Alexis Beaupré-Laperrière et al, descripts characteristics of carbonate system in the Canadian Arctic Archipelago and its adjacent Canada Basins in the recent 5 years. However, the MS needs to be majorly revised as to answer those questionnaires such as the follows: 1ïïjŐThe abstract needs to rewrite as to focus on important conclusions and avoid too much descriptive. 2ïïjŐLack of nutrient DO and other auxiliary parameters data. This is odd, because these parameters are usually obtained at the same time as

the carbonate system. This also leads to the discussion of this article is not strong. 3ïijŐAlthough this article is logically organized, it seems that there is too much information that is not important which affects the reader's reading. Specific comments: 1. Suggest change the title of 'Canadian Arctic" to "Canadian Arctic Archipelago and adjacent Basins". 2. The abstract is poorly written. There are too many summaries of the previous work, and the conclusion and discussion should be mention more. 3. The CCGS appears for the first time, giving it its full name. 4. In figure 1, blue and red lines can be used to represent the direction and scope of influence of Pacific water and Atlantic water respectively. 5. The color bar in figure 4 represents the suggested source in red for easy identification 6. In figure 5, there are only two colors of red and blue, so it is difficult to see the regional difference. It is recommended to use a variety of color gradients to distinguish. 7. Line 270 should be Fig. 6b, c. 8. What does the color difference mean in Figure 7? Suggestions clearly marked. 9. Line 325 aragonite saturation coincide with the temperature maximum but pH does not. 10. From 460 to 465 lines, this part should be an important highlight of this article, it is suggested to strengthen the discussion. 11. Line 485-495ïijŇThe author's description of the calculation method is not clear enough. My understanding is to calculate pCO2 by DIC and TA in the starting year, and then assume that the change of atmospheric CO2 is synchronized with the change of water body, calculate the DIC value only affected by the atmosphere in given year, and finally use the measured DIC minus DICatmosphere =delta DICorganism? 12. Line 520, Fig. 14 or Fig. 12ïij§ 13. Suggest moving the Appendix to method. 14. It is shocking and strange that this MS does not mention this article: Azetsu-Scott, Kumiko, Calcium carbonate saturation states in the waters of the Canadian Arctic Archipelago and the Labrador Sea JGR 2010. 15. Suggest ðİŘůðİŘijðİŘűðİŚĆðİŚŘðİŠàðİŚŠðİŚ§ðİŚčðİŚŠðİŚŠ and ðİŘůðİŘijðİŘűðİŚĚðİŚŠðİŚŞðİŚŠðİŚ§ðİŚŠðİŚŽðİŚŘðİŚŠ to simplify into ðİŘůðİŘijðİŘűðİŚĆðİŚŘðİŚă and ðİŘůðİŘijðİŘűðİŚĚðİŚŠðİŚŞ for consistency of ðİŘůðİŘijðİŘűðİŘţðİŘijðİŚĆ.

---

## Referee Comment (RC4) · Anonymous Referee #2 · 24 Mar 2020

15. Suggest DICobserved and DICreference to simplify into DICobs and DICref for consistency of DICbio.

---

## Author Comment (AC1) · 14 May 2020

Author Responses to Anonymous Referee #2

Reviewer's comments are marked as RC and author responses as AR

RC: The Arctic Ocean is particularly vulnerable to acidification due to its relatively low buffering capacity and, thus is considered as a bellwether to study global ocean acidification. The manuscript "The recent state and variability of the carbonate system of the Canadian Arctic in the context of ocean acidification" written by Alexis Beaupré-Laperrière et al, descripts characteristics of carbonate system in the Canadian Arctic

[Figure]

Archipelago and its adjacent Canada Basins in the recent 5 years. However, the MS needs to be majorly revised as to answer those questionnaires such as the follows:

RC: The abstract needs to rewrite as to focus on important conclusions and avoid too much descriptive.

AR: See response to specific comment #2.

RC: Lack of nutrient DO and other auxiliary parameters data. This is odd, because these parameters are usually obtained at the same time as the carbonate system. This also leads to the discussion of this article is not strong.

AR: The manuscript focuses on the carbonate system parameters and other inorganic parameters, which, given the spatial extent of the study area, represent a sizeable dataset and allows for a pertinent analysis within this scope.

Nutrient and dissolved oxygen data were indeed collected by other researchers in most of the oceanographic campaigns (and in some cases are available through the various data repositories listed in this manuscript). We acknowledge that a more comprehensive discussion, including correlations between DIC and AOU, could be elaborated with the inclusion of these data, and it is our hope that our manuscript will inspire other to do so.

RC: Although this article is logically organized, it seems that there is too much information that is not important which affects the reader's reading.

AR: We are unsure about what the reviewer is specifically referring to. Given the spatial extent and the complexity of hydrographic features of the study area, we believe that detailed background information is warranted.

Specific comments:

RC1: Suggest change the title of 'Canadian Arctic" to "Canadian Arctic Archipelago and adjacent Basins".

AR1: We thank the reviewer for this suggestion. The suggested formulation was in the original title of the manuscript but was changed in the interest of keeping the title short. We will apply this suggestion to the revised manuscript.

RC2: The abstract is poorly written. There are too many summaries of the previous work, and the conclusion and discussion should be mention more.

AR2: We acknowledge that the abstract contains a large amount of background information and few key results. A more complete summary of the key results will be added to the abstract. We will shorten the background information in the abstract, but we believe that some of this information is critical to most readers' understanding of the context and impetus of this study. We plan to keep this information in the abstract, albeit written more concisely.

RC3: The CCGS appears for the first time, giving it its full name.

AR3: The acronym will be defined in the revised manuscript, it stands for Canadian Coast Guard Ship.

RC4: In figure 1, blue and red lines can be used to represent the direction and scope of influence of Pacific water and Atlantic water respectively.

AR4: The (blue) arrows in figure 1 represent the dominant surface circulation flow paths, for which there is no clear distinction between Pacific and Atlantic sources, as these latter water masses only dominate at greater depths. If the reviewer insists, we could add colored lines to represent the major inflows of Pacific and Atlantic waters at depth, but these would clutter the figure. A description of the structure of the water column and the location (depth) of the Pacific and Atlantic waters masses can be found in the text of manuscript.

RC5: The color bar in figure 4 represents the suggested source in red for easy identification.

AR5: The color code in Figure 4 represents $pCO_2$ in the surface waters (< 5m). We do

not understand what the reviewer refers to by the 'source' and what he proposes. Is he referring to the surface or deep flow pattern?

RC6: In figure 5, there are only two colors of red and blue, so it is difficult to see the regional difference. It is recommended to use a variety of color gradients to distinguish.

AR6: We deemed that a divergent color scale was the most appropriate choice to distinguish between the omega-aragonite values above (supersaturated) and below 1 (undersaturated). Multi-color diverging colormaps exist, but include both red and green, colors that should be avoided for colorblind readers.

RC7: Line 270 should be Fig. 6b, c.

AR7: Figure 6 includes only one panel and therefore doesn't contain panels b or c. Nevertheless, a reference to Figure 7(k) will be added to the text, as it also shows the minima referred to in this sentence.

RC8: What does the color difference mean in Figure 7? Suggestions clearly marked.

AR8: In the panels with a legend and containing two colors (b, c, f, g, j, k), the colors are used to distinguish profiles from different sub-regions. In the panels where only one sub-region is displayed, multiple colors are used to more easily distinguish the continuity of the different profiles.

RC9: Line 325 aragonite saturation coincide with the temperature maximum but pH does not.

AR9: The passage in question reads as follows:

"The $\Omega A$ crosses the saturation threshold back to supersaturation between 200 and 250 meters, where Atlantic waters become predominant, as evidenced by a +0.8 °C temperature maximum at ~500 m (in contrast to a temperature minimum of -1.5 °C in the UHL). $\Omega A$ and $pH_T$ remain, respectively, above 1.3 and 8.05 in this layer, with respective maxima of 1.74 ($\pm$ 0.16) and 8.17 ($\pm$ 0.03) that generally coincide with the

temperature maximum mentioned previously."

For space considerations, temperature is not shown in figure 7. We instead state that the temperature maximum associated with the Atlantic water mass is found close to 500m depth, an approximation, as the core depth of specific water masses varies between stations. The maxima in $\Omega A$ and pH do coincide with each other (they are measured at the same depth, at the same station), but are closer to a depth of 400m than 500m, which is why we used the formulation "that generally coincide with the temperature maximum". The depth of the $\Omega A$ and pH maxima at this particular station will be explicitly stated in the revised manuscript, but this does not modify the stated relationship between the trends in $\Omega A$, pH and the nature of the predominant water masses.

RC10: From 460 to 465 lines, this part should be an important highlight of this article, it is suggested to strengthen the discussion.

AR10: We elected to focus the bulk of the section on the time of emergence in Appendix A, to keep the length of the article acceptable and because, as stated on line 466-467, we recognize that the dataset used to perform this statistical analysis is insufficient to insure its results are entirely reliable. With this in mind, we found it appropriate to briefly describe our results along with this important caveat in the main text, because these results are relevant to the discussion and might set an important precedent for estimating the time of emergence of the anthropogenic acidification signal using observational datasets. Nonetheless, we want to avoid putting too much emphasis on these results in the main text, as their predominance in the discussion might be misconstrued as a token of their validity.

RC11: Line 485-495 The author's description of the calculation method is not clear enough. My understanding is to calculate pCO2 by DIC and TA in the starting year, and then assume that the change of atmospheric CO2 is synchronized with the change of water body, calculate the DICvalue only affected by the atmosphere in given year, and

finally use the measured DIC minus DICatmosphere =delta DICorganism?

AR11: This issue was noted by another reviewer and will be addressed in the revised manuscript. A clearer description of the calculation procedures would be as follows: Delta-DIC-Bio is obtained by subtracting the DIC calculated at a reference time and adjusted for the atmospheric change in pCO2, from the observed DIC at the time of interest.

RC12: Line 520, Fig. 14 or Fig. 12

AR12: Figure 12. We thank the reviewer for pointing out this mistake, it will be fixed in the revised manuscript.

RC13: Suggest moving the Appendix to method.

AR13: The appendix contains some discussion and interpretation. We therefore do not consider it appropriate for the methods section.

RC14: It is shocking and strange that this MS does not mention this article: Azetsu-Scott, Kumiko, Calcium carbonate saturation states in the waters of the Canadian Arctic Archipelago and the Labrador Sea JGR 2010.

AR14: This article was indeed part of the literature review and will be added as a reference in the revised manuscript.

RC15: Suggest DICobserved and DICreference to simplify into DICobs and DICref for consistency of DICbio.

AR15: This suggestion will be applied to the revised manuscript.

---

## Author Comment (AC2) · 15 May 2020

Author Responses to Leif Anderson (Referee)

Reviewer comments are marked as RC and author responses as AR

RC0: This manuscript includes a very thorough data assessment of the carbon system parameters in Canadian Arctic waters and clearly shows the large variability both in time and space. It confirmed already published conclusions that much of Arctic shallow waters are undersaturated with respect to calcium carbonate, especially aragonite. The basis for this fact is both lowering of salinity by river runoff and sea ice melt in the

summer, as well as degradation of organic matter, both in the runoff and in the shelf seas. A valuable contribution is evaluation of "time of emergence" that shows the need for several decades of data in order to separate variability from trends.

AR0: We thank the reviewer for his encouragement and incisive comments.

RC1: Another novel approach is to compute the biological contribution to DIC time variations in the top 25 m, i.e. how much of the ocean acidification signal that is of biological activity. They find this to be about one quarter; a number that I argue has to be taken with great care. The reason is that the DIC-reference computation assumes a constant difference in air-sea $pCO_2$ with time. They mention this but do not relate this assumption to the sea ice coverage before and at the time of measurement. In this region of very temporal variation in sea ice coverage this could have a substantial impact. In the manuscript the decrease in sea ice coverage in the Canadian Basin is mentioned as a cause of elevated uptake of atmospheric $CO_2$, so the authors are aware of this aspect. One could use the satellite images to assess potential changes in the sea ice cover of the region at the times of interest, if these data have enough special and temporal coverage. Considering the fact of changing sea ice coverage I am not sure that the statement on line 504 that the calculated delta DIC-bio is beyond the uncertainty.

AR1: The reviewer points out an important fact that is not specifically mentioned in the manuscript: the uncertainty associated with the change in sea-ice cover is not explicitly stated as being a part of our assumption of constant air-sea $pCO_2$ gradient. Nonetheless, quantifying it at specific locations over different timespans is a study in of itself and entails great complexity, especially in a dynamic environment such as an archipelago: beyond simply estimating the amount of ice cover at a given location from aerial or satellite footage, one would have to account for ice advection, advection of $CO_2$ in the underlying water column, and estimate how the change in ice cover modulates air-sea gas exchange, both physically and via local chemical changes during ice formation and decay, such as with brine rejection; (see Rysgaard et al., 2011). We consider the

quantification of these processes to be fraught with uncertainties and largely beyond the scope of our study. Finally, it is worth noting that, for logistical (navigation) reasons, sampling of the water column most often took place under limited ice cover.

We thus propose to explicitly state that the change in sea-ice cover likely represents a considerable source of uncertainty on deltaDICBIO, clarify what the uncertainty we mention represents (the uncertainty propagation from the raw data analytical uncertainty and through the calculations), replace or clarify any ambiguous statements such as Âń beyond the uncertainty Âż, and caution the reader about the conclusions drawn from this quantity.

Rysgaard, S., Bendtsen, J., Delille, B., Dieckmann, G., Glud, R., Kennedy, H., Mortensen, J., Papadimitriou, S., Thomas, D. & Tison, Jean-Louis. (2011). Sea ice contribution to the air–sea CO2 exchange in the Arctic and Southern Oceans. Tellus B. 63. 10.3402/tellusb.v63i5.16409.

RC2: Further, the computation of delta DIC-bio, as described from line 480, is described in a complicated way. I had to read this section several times before understanding what had been done. My understanding is that DIC-ref is computed from TA and pH-t at the starting date, and then corrected for the increase in DIC by the increasing atmospheric pCO2, assuming a constant air-sea delta value. This could be described in a strait forward way.

AR2: The reviewer's understanding is correct and we will simplify the explanation of this parameter in the revised manuscript using a similar phrasing.

Generally this manuscript is well written, but there are several more elements that need more care.

RC3: In line 48 it reads "of these phenomena". What do these refer to?

AR3: It refers to the different chemical reactions that we collectively label as ocean acidification. This will be clarified in the revised manuscript.

RC4: Line 109 reads "and expanding undersaturation". Of what? I assume omega, but as CO2 is also discussed in this section one has to be explicit.

AR4: Correct, undersaturation of seawater with respect to aragonite. This will be clarified in the revised manuscript.

RC5: Line 165. m-Cresol purple was used for pH determination. Specify if it was purified version or not as this have significant impact on the accuracy, even when calibration against TRIS buffer. Information on the magnitude of correction of the measured value after calibration should be added.

AR5: Over the years, we have used both unpurified and purified (purchased from Prof. Robert Byrne) m-cresol purple. The stated accuracy of our spectrophotometric pH measurements is based on the reproducibility of thousands of pH measurements carried out with both m-cresol purple and phenol red, thus independent measurements, throughout most of the water column over the years. As the pH of surface waters are often higher than deeper in the water column, the absorbance ratio of the phenol red indicator in the former is high (> 3.5) and therefore outside the optimal range of its use. Hence, pH measurements in surface waters are typically based solely on m-cresol purple absorbance ratio measurements. In a recent study across a range of salinities (in a fjord), Delaigue et al. (2020) reported "The salinity-dependence of the dissociation constants and molar absorptivities of the indicators were taken from Robert-Baldo et al. (1985) for phenol red and from Clayton and Byrne (1993) for m-cresol purple. Results computed from these parameters yielded results that were more similar to each other as well as to potentiometric glass electrode measurements than the revised equation for the purified m-cresol purple provided by Douglas and Byrne (2017)."

Clayton, T. D., and Byrne, R. H.: Spectrophotometric seawater pH measurements: total hydrogen ion concentration scale calibration of m-cresol purple and at-sea results, Deep-Sea Res. Pt. I, 40, 2115-2129, 1993.

Delaigue, L., Mucci, A., and Thomas, H.: Spatial variations of the CO2 fluxes in the

Saguenay Fjord (Québec, Canada) and results of a water mixing model. Biogeosciences, 17, 547-566, 2020

Douglas, N. K., and Byrne, R. H.: Achieving accurate spectrophotometric pH measurements using unpurified meta-cresol purple, Mar. Chem., 190, 66-72, 2017.

Robert-Baldo, G. L., Morris, M. J., and Byrne, R. H.: Spectrophotometric determination of seawater pH using phenol red, Anal. Chem., 57, 2564-2567, 1985.

RC6: In line 228 DIC-bio is mentioned before it has been defined. Hence move it to the section where it is defined.

AR6: As suggested by the reviewer, this sentence will be moved to the appropriate section in the revised manuscript.

RC7: Sentence starting on line 242 does not specify depth range. I assume it refers to the surface water, and if so it should be specified.

AR7: The salinity difference mentioned in this sentence applies to most of the water column. The depth range will be explicitly stated in the revised manuscript.

RC8: Line 258. I don't see the meaning in giving mean values of here as the range is so large and the mean is very much depending on the number of samples included in the different sub-regions. This also refers to the data on line 295.

AR8: Means of aragonite and calcite saturation states are indeed dependent on the number of samples included and the spatial distribution of stations in each sub-region. Nonetheless, means inform the reader on which area has generally lower or higher surface omega values. Simply stating ranges paints an even more incomplete picture of reality, as it would be possible for most values to be close to the minimum or the maximum.

RC9: Sentence starting on line 282 is confusing. What do you rule out? That air-sea exchange has not increased the omega value, or that primary production has not?

Specify.

AR9: We rule out diurnal fluctuations of biological activity as an explanation for the omega minima we observe (the hypothesis that the lowest values would be observed during the night). This will be clarified in the revised version of the manuscript.

RC10: When it refers to Fig. 7 in the text it would be useful to also give the letters of the individual sub-figures.

AR10: This is a good suggestion; it will be applied to the revised manuscript.

RC11: Line 311. I assume that the pCO2 and pH-t values are at in situ temperature, but it would be nice to have this specified.

AR11: It will be specified in the revised manuscript.

RC12: The statements starting with the sentence on line 355 needs some more explanations. As salinity is not presented (more than as averages in the table) the reader cannot judge the statement of weaker stratification. So how does this weakening agrees with the statement on line 359 of low salinities from freshwater accumulation followed by efficient mixing? I would assume that addition of freshwater to the surface would strengthen stratification and thus hampering mixing. Please clarify.

AR12: A brief description of the salinity data will be added to this section for the Queen Maud Gulf stations to better illustrate the weaker stratification, or a figure will be added. See Figure 1 of this response for the salinity profiles. Whereas one of the salinity profiles in the area shows that stratification may be stronger because of a very low surface salinity (SP=22) due to riverine input, stratification is generally weaker in this area. The addition of freshwater to the surface and factors that result in more efficient mixing (shallow depth, stronger tidal mixing, hypothesized high residence time of water) all have competing effects on stratification, but are not mutually exclusive. Thus, we do not consider this to be a contradiction, but will address it in the revised manuscript.

RC13: Line 362. It states that the entire water column is undersaturated, but in the

figure not all are. Same goes for line 549 in conclusions. Change.

AR13: These sentences will be clarified to something akin to "The Queen Maud Gulf is the only area where, at some locations, the water column is entirely unsaturated with respect to aragonite".

RC14: Sentence starting on line 439. Such conditions would gain on being compared with the dominating atmospheric pressure fields at the time (and before) the observations.

AR14: We appreciate the suggestion but consider it outside the scope of our study that is mostly focused on water chemistry.

RC15: Suggestion ending the paragraph on line 449. I would challenge this explanation as the 2004 pCO2 data seems quite close to atmospheric levels, while the 2008 data are substantially below. Also the data at 140 m (?) show a difference between the years that is similar. The two options to get undersaturated CO2 is cooling or primary production. How do you get this signature down to 300-400 m depths? I can only see that there must be two different water masses present (if the data are correct that I do not doubt).

The vertical carbon export we called upon to explain the difference in pCO2 at depths of 300-400m between 2004 and 2008 referred to the particulate organic carbon that settles from the surface and is metabolized by bacteria at depth. After further consideration, we acknowledge that this explanation might be unlikely, as the greater pCO2 undersaturation observed near the surface (top 50m) in 2008, relative to 2004, may imply that surface primary productivity was greater in 2008. If gravitational export of biogenic carbon export was responsible for the transfer of metabolisable carbon at depth, pCO2 should be higher in 2008 at 300-400m, when the opposite is observed. The salinity and temperature data, shown in figure 2 of this response, are nearly identical in 2004 and 2008, which is not consistent with a shift in water masses/stratification or cooling. Alkalinity values at 300-400m depth in 2008 are abnormally high, which

might explain the low pCO2 values. Nonetheless, a verification of the specific alkalinity at similar depths in surrounding stations revealed no systematic error. Therefore, we currently cannot propose a plausible explanation for the low pCO2 values observed at this station in 2008. We will outline this in the revised manuscript.

RC16: Line 464. Make 2 and A subscript in pCO2 and omega A AR16: This will be modified in the revised manuscript.

RC17: Line 608. The statement that the interannual variability should be relatively small at these depths is not necessarily true. In this dynamic region the contribution of different water masses might vary significantly also at these depths.

AR17: True; a caveat will be added to this sentence. The point made in this sentence nonetheless remains, that is, interannual variability is expected to be smaller at these depths relative to the surface.

RC18: Finally I was surprised that there is no reference to the below article that is highly relevant to this contribution. Azetsu-Scott, K., A. Clarke, K. Falkner, J. Hamilton, E.P. Jones, C. Lee, B. Petrie, S. Prinsenberg, M. Starr and P. Yeats, 2010. Calcium carbonate saturation states in the waters of the Canadian Arctic Archipelago and the Labrador Sea. Journal of Geophysical Research, 115:C11021, doi:10.1029/2009JC005917.

AR18: This article was part of the literature review and should indeed be cited in the article. It will be cited in section 1, where we describe the state of knowledge on carbonate mineral saturation states in the study area.

[Figure]

**Fig. 1.** Salinity profiles in the central CAA (Canadian Arctic Archipelago) and QMG (Queen Maud Gulf)

[Figure]

**Fig. 2.** Salinity, Total alkalinity (umol/kg) and Temperature (degrees C) profiles at station AM1

---

## Author Response (AR1)

**Author response to Associate Editor Decision: Publish subject to minor revisions (review by editor) (22 May 2020)**

We are submitting, along with this document, a revised version of the manuscript following our replies to the referees' comments, and a version of the same document with the modified portion of the text written in red font. The modifications follow exactly the modifications proposed as responses to the reviewers, which have not changed.

Below is a list of the reviewer comments which have resulted in a change in the manuscript, along with the line number of where the changes are found in the revised manuscript. Please refer to our responses if the nature of the change is unclear.

Following the editor's request regarding dataset availability, the full dataset of primary data (total alkalinity, pHt, dissolved inorganic carbon) has been uploaded to the Polar Data Catalogue, where it is currently awaiting approval. The new file will appear as part of the same entry as Mucci (2017) shortly. We will send a note to the Associate Editor when it becomes available.

**Leif Anderson**

- RC1: Another novel approach is to compute the biological contribution to DIC time variations in the top 25 m, i.e. how much of the ocean acidification signal that is of biological activity. They find this to be about one quarter; a number that I argue has to be taken with great care. The reason is that the DIC-reference computation assumes a constant difference in air-sea pCO2 with time. They mention this but do not relate this assumption to the sea ice coverage before and at the time of measurement. In this region of very temporal variation in sea ice coverage this could have a substantial impact. In the manuscript the decrease in sea ice coverage in the Canadian Basin is mentioned as a cause of elevated uptake of atmospheric CO2, so the authors are aware of this aspect. One could use the satellite images to assess potential changes in the sea ice coverage. Considering the fact of changing sea ice coverage I am not sure that the statement on line 504 that the calculated delta DIC-bio is beyond the uncertainty. (LINES 511-513, 517)
- RC2: Further, the computation of delta DIC-bio, as described from line 480, is described in a complicated way. I had to read this section several times before understanding what had been done. My understanding is that DIC-ref is computed from TA and pH-t at the starting date, and then corrected for the increase in DIC by the increasing atmospheric pCO2, assuming a constant air-sea delta value. This could be described in a strait forward way. (LINES 490-494)
- RC3: In line 48 it reads "of these phenomena". What do these refer to? (LINE 49)
- RC4: Line 109 reads "and expanding undersaturation". Of what? I assume omega, but as CO2 is also discussed in this section one has to be explicit. (LINE 113)
- RC6: In line 228 DIC-bio is mentioned before it has been defined. Hence move it to the section where it is defined. (MOVED TO LINES 509-510)
- RC7: Sentence starting on line 242 does not specify depth range. I assume it refers to the surface water, and if so it should be specified. (LINE 245)
- RC9: Sentence starting on line 282 is confusing. What do you rule out? That air-sea exchange has not increased the omega value, or that primary production has not? Specify. (LINES 284-287)
- RC10: When it refers to Fig. 7 in the text it would be useful to also give the letters of the individual sub-figures. (MULTIPLE LINES)

- RC11: Line 311. I assume that the pCO2 and pH-t values are at in situ temperature, but it would be nice to have this specified. (LINE 315)
- RC12: The statements starting with the sentence on line 355 needs some more explanations. As salinity is not presented (more than as averages in the table) the reader cannot judge the statement of weaker stratification. So how does this weakening agrees with the statement on line 359 of low salinities from freshwater accumulation followed by efficient mixing? I would assume that addition of freshwater to the surface would strengthen stratification and thus hampering mixing. Please clarify. (LINES 358-362)
- RC13: Line 362. It states that the entire water column is undersaturated, but in the figure not all are. Same goes for line 549 in conclusions. Change. (ABSTRACT, 367, 563)
- RC15: Suggestion ending the paragraph on line 449. I would challenge this explanation as the 2004 pCO2 data seems quite close to atmospheric levels, while the 2008 data are substantially below. Also the data at 140 m (?) show a difference between the years that is similar. The two options to get undersaturated CO2 is cooling or primary production. How do you get this signature down to 300-400 m depths? I can only see that there must be two different water masses present (if the data are correct that I do not doubt). (LINES 454-459)
- RC16: Line 464. Make 2 and A subscript in pCO2 and omega A (LINE 473)
- RC17: Line 608. The statement that the interannual variability should be relatively small at these depths is not necessarily true. In this dynamic region the contribution of different water masses might vary significantly also at these depths. (LINES 613-614)
- RC18: Finally I was surprised that there is no reference to the below article that is highly relevant to this contribution. Azetsu-Scott, K., A. Clarke, K. Falkner, J. Hamilton, E.P. Jones, C. Lee, B. Petrie, S. Prinsenberg, M. Starr and P. Yeats, 2010. Calcium carbonate saturation states in the waters of the Canadian Arctic Archipelago and the Labrador Sea. Journal of Geophysical Research, 115:C11021, doi:10.1029/2009JC005917. (LINES 76-81)

**Reviewer 2**

- RC1: Suggest change the title of 'Canadian Arctic" to "Canadian Arctic Archipelago and adjacent Basins". (TITLE)
- RC2: The abstract is poorly written. There are too many summaries (ABSTRACT) of the previous work, and the conclusion and discussion should be mention more.
- RC3: The CCGS appears for the first time, giving it its full name. (LINE 152)
- RC7: Line 270 should be Fig. 6b, c. (LINE 272)
- RC9: Line 325 aragonite saturation coincide with the temperature maximum but pH does not. (LINE 327-332)
- RC11: Line 485-495 The author's description of the calculation method is not clear enough. My understanding is to calculate pCO2 by DIC and TA in the starting year, and then assume that the change of atmospheric CO2 is synchronized with the change of water body, calculate the DICvalue only affected by the atmosphere in given year, and finally use the measured DIC minus DICatmosphere =delta DICorganism? (LINES 490-494)
- RC12: Line 520, Fig. 14 or Fig. 12 (LINE 532)
- RC14: It is shocking and strange that this MS does not mention this article: Azetsu-Scott, Kumiko, Calcium carbonate saturation states in the waters of the Canadian Arctic Archipelago and the Labrador Sea JGR 2010. (LINES 76-81)
- RC15: Suggest DICobserved and DICreference to simplify into DICobs and DICref for consistency of DICbio. (LINE 489 AND FOLLOWING PARAGRAPH)

[revised manuscript text omitted]
   | 30.96 | 29.69 - 32.10 | 2138                        | 2062 - 2209 | 1999            | 1920 - 2058 |

---

## Author Response (AR2)

**Author response to Associate Editor Decision: Publish subject to technical corrections (24 Jun 2020)**

The modifications, all except the last suggested by the editor, were made in the submitted manuscript :

- « in-situ » was changed to « in situ » in the entire manuscript

- « meters » was changed to « m » in the entire manuscript

- Line 190 : Mentions of CO2SYS as an algorithm were replaced by « software » (line 190)

- Line 494 : The availability of the data from Dlugokencky and Tans was clarified by providing the URL to the repository website (line 494)

- Line 192 : The expression $\partial pCO2 / \partial T / pCO2$ was replaced by its correct mathematical form $\partial \ln(pCO2) / \partial T$, which is the one mentioned most often in the literature (including in Dinauer and Mucci, 2017, Biogeosciences, https://doi.org/10.5194/bg-14-3221-2017).

- Addition of grid and uniformisation of coordinates to certain figures.